# Microsurgery Robots: Applications, Design, and Development

**DOI:** 10.3390/s23208503

**Published:** 2023-10-16

**Authors:** Tiexin Wang, Haoyu Li, Tanhong Pu, Liangjing Yang

**Affiliations:** 1ZJU-UIUC Institute, International Campus, Zhejiang University, Haining 314400, China; wangtiexin@zju.edu.cn (T.W.); haoyu.21@intl.zju.edu.cn (H.L.); tanhong.21@intl.zju.edu.cn (T.P.); 2School of Mechanical Engineering, Zhejiang University, Hangzhou 310058, China; 3Department of Mechanical Engineering, University of Illinois Urbana-Champaign, Urbana, IL 61801, USA

**Keywords:** microsurgery robot (MSR), mechanism design, imaging and sensing, control and automation, human–machine interaction (HMI)

## Abstract

Microsurgical techniques have been widely utilized in various surgical specialties, such as ophthalmology, neurosurgery, and otolaryngology, which require intricate and precise surgical tool manipulation on a small scale. In microsurgery, operations on delicate vessels or tissues require high standards in surgeons’ skills. This exceptionally high requirement in skills leads to a steep learning curve and lengthy training before the surgeons can perform microsurgical procedures with quality outcomes. The microsurgery robot (MSR), which can improve surgeons’ operation skills through various functions, has received extensive research attention in the past three decades. There have been many review papers summarizing the research on MSR for specific surgical specialties. However, an in-depth review of the relevant technologies used in MSR systems is limited in the literature. This review details the technical challenges in microsurgery, and systematically summarizes the key technologies in MSR with a developmental perspective from the basic structural mechanism design, to the perception and human–machine interaction methods, and further to the ability in achieving a certain level of autonomy. By presenting and comparing the methods and technologies in this cutting-edge research, this paper aims to provide readers with a comprehensive understanding of the current state of MSR research and identify potential directions for future development in MSR.

## 1. Introduction

Microsurgery is a surgical procedure that involves operating on small structures with the aid of a surgical microscope or other magnifying instrument. The visual magnification allows surgeons to operate on delicate structures with greater precision and accuracy, resulting in better treatment outcomes. With its unique advantages, microsurgery has been widely adopted in various surgical specialties, including ophthalmology, otolaryngology, neurosurgery, reconstructive surgery, and urology, where intricate and precise surgical tool manipulation on a small scale is required [1].

Despite its benefits, microsurgery also presents significant challenges. Firstly, the small and delicate targets in microsurgery require a high level of precision, where even the slightest tremor may cause unnecessary injury [2]. Microsurgery requires complex operations under limited sensory conditions, such as limited microscope field of view and low tool–tissue interaction force [3]. This is demanding on the surgeon’s surgical skills and requires extensive training before the surgeon can perform such surgical procedures clinically. In addition, microsurgery often requires the surgeon to perform prolonged tasks in uncomfortable positions, which can lead to fatigue and increase the risk of inadvertent error [4].

To address these challenges, a growing number of researchers have begun to explore the use of robotic technologies in microsurgery and have developed various microsurgery robotic (MSR) systems. The MSR system has the potential to make a significant impact in the field of microsurgery. It can provide surgeons with increased precision and stability by functions such as tremor filtering and motion scaling [5]. By integrating various perception and feedback technologies, it can provide richer information about the surgical environment and offer intuitive intraoperative guidance [4,6]. It can also enhance surgeons’ comfort during surgical operations through ergonomic design. With the above features, surgeons can improve their surgical performance with the help of MSR and even accomplish previously impossible surgical tasks [7].

Figure 1 counts the number of MSR-related articles published per year from 2000 to 2022, and its data were obtained by searching different keywords in Google Scholar. As can be seen from the figure, there is an overall upward trend in the number of relevant studies on MSR.

To date, a large number of reviews have been published discussing the application of MSR systems in specific surgical specialties. Tan et al. introduced the current state of robotic technology in reconstructive microsurgery and examined the barriers to widespread adoption of surgical robots in this field [8]. Aitzetmüller et al. described the development of surgical robotic systems in the field of reconstructive plastic surgery and then presented several types of corresponding MSR systems and their potential for integration into clinical practice [9]. Vander Poorten et al. provided a comprehensive review of MSR systems for retinal surgery, covering state-of-the-art robotic systems as well as advances in sensing, modeling, visualization, and control [10]. Smith et al. discussed the history and current status of neurosurgical robotic systems, highlighting their applications in microsurgery, stereotaxy, and radiotherapy, and outlined future directions for robotic neurosurgery [11]. Parekattil and Moran reviewed the development of robotic devices in urologic surgery and discussed the various applications of MSR robots in urologic surgery [8]. Similarly, Gudeloglu et al. reviewed the current status of robotic microsurgery in male infertility and urologic surgery [12].

Compared to the reviews that summarize the research on MSR for specific surgical specialties, there is a limited amount of literature that provides an in-depth review of the relevant technologies used in MSR systems. Mattos et al. discussed the benefits of MSR systems for surgeons and outlined the current challenges in robotic microsurgery from the perspectives of mechatronics, perception, and the surgeon-robot interface [1]. They also presented a case study of the relevant technologies of an MSR system called RALP for laser phonomicrosurgery. Zhang et al. reviewed the development of MSR systems from teleoperation to automation [13]. They categorized existing MSR systems based on the grounding of the robot, and provided a detailed description of the key features of the ungrounded systems. In addition, they discussed the application of learning-based approaches to achieve the automation of robots.

Despite these valuable contributions, the above papers remain insufficient to comprehensively represent the related technologies of MSR systems. Therefore, the purpose of this article is to provide an in-depth technical introduction to MSR systems to help readers gain a comprehensive understanding of the current state of research in MSR and to identify potential directions for future development in this cutting-edge field.

This review article is structured as follows: Section 2 provides an overview of the challenges faced in different specialties of microsurgery with examples. Section 3 details the MSR-related technologies used to address these challenges, including the robot design, perception technologies, human–machine interaction (HMI), and automation. In Section 4, the characteristics and main parameters of various classic MSR systems are introduced in detail from a more specific point of view. Finally, the current challenges and future directions of the MSR are presented in Section 5, and a concise summary of the article is provided in Section 6.

## 2. Challenges of Microsurgery in Various Surgical Specialties

Microsurgery has revolutionized several surgical specialties with its high precision and meticulous techniques. However, despite its many benefits, microsurgery also presents unique challenges in various surgical departments. The purpose of this chapter is to summarize and analyze the specific difficulties encountered in various surgical specialties when implementing microsurgical procedures. By identifying these challenges, we can better understand the requirements and potential areas for improvement of microsurgery in different settings. In this chapter, we will sequentially introduce the challenges of microsurgery in five different specialties, including ophthalmology, otolaryngology, neurosurgery, reconstructive surgery, and urology.

### 2.1. Ophthalmology

Ophthalmology is a branch of medicine that focuses on the structure, function, and diseases of the eye. Due to the small target and limited operating space of the eye, manual ophthalmic surgery requires highly precise and dexterous manipulation under the surgical microscope, which can be challenging even for experienced surgeons. In addition to general ophthalmic procedures such as cataract surgery and glaucoma surgery, there are several procedures that are particularly difficult to perform manually and could benefit from robotic assistance. These include epiretinal membrane peeling, retinal vein cannulation (RVC), and subretinal injections [10].

Epiretinal membrane (ERM) peeling is a treatment for retinal wrinkling and distortion caused by shrinkage forces applied to the macular region, which can lead to decreased visual acuity and metamorphopsia. This procedure is an essential therapy for conditions such as macular traction syndrome, macular hole, macular folds, and advanced diabetic retinopathy. The treatment involves vitrectomy pars plana with subsequent excision of the ERM and removal of the internal limiting membrane (ILM), which has an average thickness of 2.5 µm [14,15]. This process aims to reduce the recurrence of ERM [16,17]. During ERM peeling, surgeons must carefully remove an average of 61 ± 28 µm of epiretinal membrane, while minimizing complications [18]. Complication rates range from 2% to 30%, primarily due to the lack of depth information, which can result in tool misplacement and hemorrhage or even retinal damage.

Retinal vein occlusion (RVO) is the most common retinal vascular disease resulting from thrombus formation, hypercoagulability, or other causes of reduced blood flow in the retinal vein [19,20,21]. Traditional treatments for RVO, such as radial optic neurotomy, laser photocoagulation, hemodilution, and intravitreal injections, can only prevent disease progression and are not curative [7]. Retinal vein cannulation (RVC) is a potential treatment that can dissolve the occlusion and restore blood flow by directly injecting an anticoagulant into the obstructed vein [22]. However, the procedure requires inserting a thin cannula (∅ 70–110 μm) into small branch retinal veins (typical ∅ < 200 μm) during RVC, a task made difficult by the human hand tremor (RMS amplitude around 182 µm [23]) and the imperceptible tool–tissue interaction forces [24].

Subretinal injection is an intravitreal drug delivery method increasingly used in ophthalmology for the treatment of common retinal diseases, such as gene and stem cell therapy, sub-macular hemorrhage, and age-related macular degeneration (AMD) [25,26]. Compared to the slow-acting therapeutic approaches of intravitreal injections, subretinal injections provide direct action on retinal tissue and precise intraocular targeting, reducing intraocular drug diffusion while maximizing the duration of interaction with the retina [27]. However, subretinal injection places high demands on the surgeon’s operations. For example, treating AMD requires the surgeon to penetrate the retina, which has a foveal thickness of less than 250 µm [28], with a microcannula and to target critical layers in the retinal pigment epithelium, which requires a precision of about 25 μm [26,29]. In addition, surgeons must carefully consider the distance of action and residence time of the drug to maximize the effect with the smallest amount of the drug.

### 2.2. Otolaryngology

Otolaryngology is the medical specialty that focuses on the diagnosis and treatment of disorders related to the ears, nose, and throat. Surgeons typically use a microscope or endoscope to visualize the surgical area, and manipulate different surgical instruments for precise tissue manipulation. Common surgical procedures in this field include cochlear implant surgery (CIS), endonasal skull base surgery (ESBS), among others. However, the complex surgical environment and inherent limitations of the human sensory-motor capabilities make these procedures challenging to perform manually.

CIS is a procedure in which surgeons implant electrode arrays into the spiral cochlea after performing a mastoidectomy, with the potential to provide hearing to patients with severe hearing impairment. In the mastoidectomy step, surgeons need to remove the bone between the surface and the inner ear to gain access to the cochlea. This requires high-precision drilling to avoid inadvertent damage to important structures while exposing anatomical structures [30]. During the insertion phase of CIS, the surgeon is faced with a significant challenge as he has no visual feedback once the implant enters the cochlea. In addition, the insertion force needs to be less than 0.1 N and the placement accuracy of the cochleostomy has to be about 0.5 mm [31]. These stringent accuracy requirements push the limits of human sensory-motor capacity, which are bounded by inevitable physiological tremors, poor dexterity, and positioning precision. These constraints make it quite challenging to implement CIS manually.

ESBS is another otolaryngologic procedure used to treat inaccessible pituitary tumors located deep within the skull and in areas with non-intuitive information about the lesion. The surgeon creates a narrow tunnel from the nasal cavity to the lesion by removing bony structures using instruments such as suction and incision instruments, bone drills, and blunt dissection instruments. Subsequently, a minimally invasive tool with a length of at least 100 mm is utilized to approach and treat the lesion [32]. The surgeon needs a microscope or endoscope for visualization and instrument guidance in ESBS [30,33]. The challenge of ESBS lies in precisely removing tumors in narrow passages while addressing barriers such as limited working space, instrument collisions, poor sensory feedback, and the amplified hand tremor when handling long surgical tools.

### 2.3. Neurosurgery

Neurosurgery is the medical specialty that focuses on the surgical intervention of disorders which affect any part of the nervous system including the brain, spinal cord, and peripheral nerves [34].

With the implementation of imaging techniques such as Magnetic Resonance Imaging (MRI) and Computed Tomography (CT), as well as stereotactic techniques in neurosurgery, surgeons can precisely locate the target brain tissue and perform tasks such as biopsies and radiotherapy [11]. But many neurosurgical procedures still require surgeons to perform precise manipulations of the tissue under a microscope. Due to the complex and delicate neural structures and the limited operating space [35,36], the neurosurgery places high demands on the surgeon’s precision, dexterity, and stamina [37].

Taking surgical interventions for glioma as an example, glioma is a common type of tumor that originates in the brain and usually grows infiltratively, with no obvious border with normal brain tissue [38]. During the surgery, the surgeon needs to operate instruments such as bipolar forceps and a suction tool under a microscope to perform precise operations on the delicate brain tissue and to remove the tumor as completely as possible while maintaining the patient’s quality of life [39]. In addition, surgeons often endure prolonged periods of uncomfortable posture to access the surgical corridor, which can lead to surgeon fatigue and inadvertent errors [4].

The superficial temporal artery to middle cerebral artery (STA–MCA) anastomosis is another neurosurgical procedure indicated for cerebral diseases such as Moyamoya disease and cerebral aneurysms. The STA-MCA anastomosis is performed in a relatively superficial region of the brain and it has been described as one of the most difficult microsurgical procedures since it involves blood vessels of 1 mm or smaller in diameter [40].

### 2.4. Reconstructive Surgery

Reconstructive surgery is a subset of plastic surgery, which is devoted to regaining the form and functionality of different body components [41]. Common reconstructive procedures include hand surgery, maxillofacial surgery, tumor removal, etc. Reconstructive surgery involves the anastomosis of tiny blood vessels, lymph, and nerves, and surgeons need to operate the needle holder, forceps, microscissors, or other instruments to perform prolonged operations on tiny targets, which requires a high degree of precision in the surgeon’s operation. The following paragraphs will introduce challenges in reconstructive surgery through specific procedures.

For autologous breast reconstruction, the deep inferior epigastric perforator (DIEP) flap method has become widely accepted [42]. This surgical procedure involves the transplantation of a flap from the patient’s abdomen to the breast. The blood vessels and nerves are transplanted along with the tissue and reconnected at the defective site, and the quality of the procedure is highly dependent on the precision of vessel reconnection. Free flaps for DIEP breast reconstruction typically choose vessels with diameters of 0.8 to 2.5 mm [41]. Larger vessels are generally found deeper in the body, resulting in a greater deficit at the donor site. Selecting finer vessels that are close to the skin can reduce the required invasiveness, but it will make the anastomosis process challenging, mainly because of the surgeon’s tremor at this scale [43].

Another reconstructive procedure is the lympho-venous anastomosis (LVA), which refers to the surgical connection of lymphatic vessels and veins. This procedure is typically used in the treatment of lymphedema, a condition in which lymph fluid accumulates in the tissues. The main challenge of LVA is the small size of the lymphatic vessels, which range from 0.3 to 0.8 mm in diameter, and requires a high level of precision and skill to suture the vessels together without causing damage or leakage [44]. Additionally, the lymphatic vessels can be difficult to identify due to their translucent appearance, small size, and location in a background of connective tissue fibers, nerves, and fibrosis [45].

In cases of fingertip or facial reconstruction, super-microsurgical techniques are required to reconnect nerves and vessels with a diameter of 0.3–0.8 mm, which are extremely difficult to perform manually due to the small size and delicate nature of the target [41].

### 2.5. Urology

Urology surgery is a surgical subspecialty that focuses on the diagnosis and treatment of conditions and disorders of the urinary-tract system, as well as the male reproductive system. Some urology surgery procedures, such as vasovasostomy (VV), vasoepididymostomy (VE), and varicocelectomy, involve delicate tissues like the vas deferens, blood vessels, and nerves [46]. These procedures need to be performed by skilled surgeons using microsurgical techniques.

Take VE as an example; it is a surgical procedure used to restore male fertility. During the procedure, the surgeon uses instruments such as microscissors and needle holders to perform a precise, tension-free anastomosis between the vas (with a luminal diameter of 0.3–0.4 mm) and the epididymal tubule (with a diameter of 0.15–0.25 mm), so that the sperm can successfully pass through [47]. Currently, three distinct intussusception techniques have been utilized: end-to-end, end-to-side, and end-to-side. This surgical procedure requires exceptional microsurgical technique and high magnification, and should only be performed by experienced microsurgeons due to its complexity [12].

### 2.6. Summary

Table 1 provides a comprehensive summary of the technical parameters and challenges associated with microsurgical tasks in different specialties.

The challenges in microsurgery can be mainly categorized into the following aspects:Microsurgery involves the manipulation of micron-scale targets, including the treatment of delicate and fragile tissues (such as epiretinal membranes or brain tissues), as well as suturing or injecting small vessels, nerves, and lymphatic channels. These tasks require a high degree of precision, and inadvertent tremors as the surgeon manipulates the instruments can reduce accuracy and potentially damage the targets;The surgeon’s perception of the surgical environment is limited in microsurgery. The limited field of view and depth of field of the surgical microscope makes it difficult to perceive the position and depth information of small or transparent targets, and the subtle interaction forces during the surgical procedure can sometimes go unnoticed by the surgeon;Microsurgical procedures require surgeons to maintain a high level of concentration, often performing prolonged surgical tasks in ergonomically unfavorable positions. This can lead to physical and mental fatigue, increasing the risk of inadvertent errors;Due to the precision and complexity of microsurgical tasks, surgeons require extensive professional training before performing clinical procedures.

These challenges impede optimal outcomes of microsurgical procedures and their widespread adoption. The MSR system has great potential to expand the field of microsurgery, which provides increased stability and safety, mitigating problems such as drift, tremor, and reduced interactive forces on affected tissues [48,49]. In several studies focusing on robotic-assisted vascular and lymphatic anastomosis, comparing the learning curves of manual operation versus robot-assisted surgery, MSR-based surgery took a longer time, but had greater microsurgical precision than traditional anastomosis methods [50,51]. However, it is worth noting that the learning curve for robotic surgery indicates a faster rate of reduction in operation time over a shorter number of trials, with both mode of operations taking comparable amounts of time to complete the task at the final stage of the learning curve [1,2]. Currently, researchers have explored the use of robotic technology in microsurgery and proposed numerous MSR systems, which will be discussed in detail in the following chapters.

## 3. Key Technologies of the MSR Systems

### 3.1. Concept of Robotics in Microsurgery

To solve the current challenges in microsurgery discussed in Section 2, more and more researchers are turning their attention to robotics. Robotics has a wide application space and great potential in the field of microsurgery, and we will introduce the technologies related to MSR in this chapter. It is believed that through the detailed introduction of these technologies, readers can gain a more comprehensive understanding of how MSR systems work, and it can provide valuable references for future research.

The typical workflow of the MSR system can be summarized in Figure 2, where the gray boxes and blue arrows indicate the elements and processes that are involved in most MSR systems, respectively, while the green arrows indicate additional functions available in some systems. During robotic surgery, the surgeon can directly or remotely transmit control commands to the MSR, which in turn operates the surgical instruments to interact with the target tissues with improved precision and stability. At the same time, surgical information is collected by the microscope and other sensors in real-time, and then transmitted to the surgeon in the form of visual feedback.

In some MSR systems, the robot can achieve environmental perception with the help of various sensors, and the information that can help improve surgical outcomes (e.g., tool–tissue interaction force, tool tip depth, danger zones, etc.) is fed back to the surgeon through the human-machine interface in different forms. Furthermore, some MSR systems have achieved a certain level of autonomy based on adequate environmental perception, allowing the robot to perform certain tasks autonomously under the supervision of the surgeon.

Based on the workflow of the MSR system illustrated in Figure 2, this chapter divides the key technologies of the MSR into four sections, which will be introduced in subsequent sections as follows:➀Operation modes and mechanism designs: As the foundation of the robot system, this section discusses the structural design and control types of the MSR;➁Sensing and perception: This is the medium through which surgeons and robots perceive the surgical environment, and this section discusses the techniques that use MSR systems to collect environmental data;➂Human–machine interaction (HMI): This section focuses on the interaction and collaboration between the surgeon and MSR, discussing techniques that can improve surgeon precision and comfort, as well as provide more intuitive feedback on surgical information;➃Automation: This section discusses technologies for the robot to automatically or semi-automatically perform surgical tasks, which can improve surgical efficiency and reduce the workload of the surgeon.

### 3.2. MSR Operation Modes and Mechanism Designs

MSR can be broadly classified as handheld, teleoperated, co-manipulated, and partially automated robots based on the control method. All of them use a microscope and/or OCT as visual feedback. In traditional surgery, the surgeon controls surgical tools with a microscope as visual feedback. MSR brings great convenience to the surgery, and the surgeon no longer needs to perform the operation completely manually. There is a system of categorizing according to the degree of robotic control from low to high, as shown in Figure 3, including handheld, teleoperated, co-manipulated, and partially automated robots.

(a) Handheld robot. In the handheld robotic system, the surgical tool itself is retrofitted into a miniature robotic system called a robotic tool. The surgeon manipulates it to perform the surgical procedure. The robotic tool provides tremor elimination, depth locking, and other functions. “Micron” is a typical example [52].

(b) Teleoperated robot. In the teleoperated robotic system, the surgeon manipulates the master module to control the slave module, which replaces the surgeon’s hand to manipulate the surgical tool. The system integrates the functions of motion scaling and tremor filtering through servo algorithms. In addition, it achieves three-dimensional perception with the integration of haptic feedback or depth perception algorithms at the end of the surgical tool. A typical example is “Preceyes Surgical System” [53,54].

(c) Co-manipulated robot. In the co-manipulated robotic system, the surgeon manipulates the surgical tool simultaneously with the robot. The surgeon manually manipulates the surgical tool directly to control the motion. At the same time, the robot also holds the surgical tool, which provides assistive compensation for hand tremor and allows for prolonged immobilization of the surgical tool. “Co-manipulator” is a typical example [22].

(d) Partially automated robot. In the partially automated robotic system, specific procedures or steps of procedures are performed automatically by the robot. The robot directly manipulates and controls the motion of the surgical tool. The processed image information is provided to the robot as feedback and guidance. Simultaneous visual information is transmitted to the surgeon, who can provide override orders to supervise the partially automated procedure at any time. “IRISS” is a typical example [55].

MSR are broadly classified into RCM mechanisms and non-RCM mechanisms based on the robot structure. Taylor et al. first introduced the concept of remote center of motion (RCM) for motion control in 1995, which restrains the movement of the surgical tools within the surgical incision [56]. Therefore, depending on the different surgical environment, for example, closed or open surgical environment, MSR adopts an RCM structure or non-RCM structure, respectively. Figure 4 shows the typical different structures of MSR.

Open surgical environments such as neurosurgery mostly utilize non-RCM mechanism robots. This type of structure is characterized by flexibility and is not limited by the shape of the mechanism. And the serial robotic arm can be directly used as the MSR’s manipulator.

Enclosed surgical environments, in the example of ophthalmic surgery, mostly utilize RCM mechanism robots. The RCM mechanism is divided into passive joint, active control, and mechanical constraint according to the implementation method.

Passive-joint RCM mechanism.

It is generally composed of two degrees of freedom vertically intersecting rotary joints, which achieve RCM through the active joint movement of the robotic arm under the restriction of the incision on the patient’s body surface. This design guarantees safety while reducing number of joints and mechanism size. However, it is easily influenced by the flexibility at the human incision and it is difficult to determine the accurate position of instrument insertion, which has a bad impact on the manipulation of precision. “MicroHand S” is a typical example [63].

Active-control RCM mechanism.

In this type of RCM mechanism, the RCM of the surgical tools around the incision is generally achieved by a programable control approach, which is usually called virtual RCM. This design is simple in structure and flexible in form. However, the precision of the movement of the surgical tools depends on the stability of the precision of the control system. The general concern is that the security of the system is guaranteed by the algorithm. “RAMS” is a typical example [68].

Mechanical-constraint RCM mechanism.

It generally used a specific mechanical mechanism to achieve the RCM of surgical tools, which is characterized by high safety under mechanical restraint. The commonly used mechanical-constraint RCM mechanisms are usually divided into four categories, namely triangular, arc, spherical, and parallelogram structures, according to the different mechanical structures. Triangular RCM mechanisms have the features of high structural stiffness but significant joint coupling [69]. The recent research nowadays proposes a novel dual delta structure (DDS), which can interchange wide motion and precise motion by adjusting the work area and precision to meet the requirements of intraocular surgery and reconstructive surgery. However, the disadvantages of the triangular mechanism are still obvious [61]. The arc RCM mechanism has a simple structure but a large volume, which is difficult to drive [60]. The spherical RCM mechanism is very compact but with poor stability and is prone to motion interference [59]. Among them, the parallelogram RCM mechanism is the most widely utilized because of its apparent advantages. Various researchers have tried different structural designs based on parallelograms in anticipation of improvements. Integrating the synchronous belt into the parallelogram mechanism can simplify the mechanism linkage and compact the structure [58]. Some researchers have utilized the six-bar structure to be capable of further improving the structural stability and increasing the diversity of the drive [70].

### 3.3. Sensing and Perception in MSR

#### 3.3.1. Imaging Modalities

Imaging technologies play a crucial role in MSR systems. In traditional microsurgery, surgeons rely primarily on the operating microscope to observe the surgical environment, which helps surgeons clearly observe the tissue and perform precise surgery by magnifying the surgical site. As technology continues to evolve, an increasing number of imaging techniques are being used in MSR systems to provide additional information about the surgical environment. These imaging techniques can help surgeons better identify the target tissue or guide the intraoperative movements of the robot, so as to achieve better surgical outcomes. This section will introduce imaging techniques with current or potential applications in MSR.

Magnetic resonance imaging and computed tomography

Magnetic resonance imaging (MRI) and computed tomography (CT) are medical imaging technologies that can generate high-resolution 3D images of the organs. MRI uses magnetic fields and radio waves to produce images with a spatial resolution of about 1 mm [71]. CT, on the other hand, uses X-rays and computer processing to create higher spatial resolution images at the sub-millimeter level [72]. Both have been widely used in preoperative and postoperative diagnosis and surgical planning.

Some researchers have attempted to apply MRI technology to robot systems to achieve intraoperative image guidance. Currently, a number of MRI-guided robot systems have been developed for stereotactic [73,74] or percutaneous intervention [75,76]. In the microsurgery field, Sutherland et al. designed an MRI compatible MSR system for neurosurgery [77]; the robotic arm is made of non-magnetic materials, such as titanium and polyetheretherketone, to prevent magnetic fields or gradients from affecting its performance and to ensure that the robot does not degrade image quality significantly. The system enables alignment of the robot arm position with the intraoperative MRI scans, allowing stereotaxy to be performed within the imaging environment (magnet bore) and microsurgery outside the magnet [37,65]. Fang et al. [78] proposed a soft robot system with 5-DOF and small size (Ø12 × 100 mm) for MRI-guided transoral laser microsurgery, and the intraoperative thermal diffusion and tissue ablation margin are monitored by magnetic resonance thermometry.

Optical coherence tomography

Optical coherence tomography (OCT) is a non-invasive imaging technique that uses low-coherence light to produce 2D and 3D images within light-scattering media. It is capable of capturing high resolution (5–20 μm) images in real-time [79] as well as visualizing the surgical instruments and subsurface anatomy, and it has gained wide application in ophthalmology [80,81]. Many researchers have integrated the OCT technology with MSR systems to provide intraoperative guidance in different dimensions (A-Scan for 1-D depth information, B-Scan for 2-D cross-sectional information, C-Scan for 3-D information) for better surgical outcomes.

Cheon et al. integrated the OCT fiber into a handheld MSR with piezo motor, and the feedback data from the OCT A-Scan was used to achieve an active depth locking function, which effectively improved the stability of the surgeon’s grasping actions [82,83]. Yu et al. designed the OCT-forceps, a surgical forceps with OCT B-Scan functionality, which was installed on a teleoperate MSR to enable OCT-guided epiretinal membrane peeling [84]. Gerber et al. combined the MSR with a commercially available OCT scanning device (Telesto II-1060LR, Thorlabs, NJ, USA), enabling the intraoperative ocular tissue identification and tool tip visual servoing, and completed several OCT-guided robotic surgical tasks, including semi-automated cataract removal [55], automated posterior capsule polishing [85], and automated retinal vein cannulation [22].

Near-infrared fluorescence

Near-infrared fluorescence (NIRF) imaging is a non-invasive imaging technique that utilizes near-infrared (NIR) light and NIR fluorophores to visualize biological structures and processes in vivo [86], and it can provide images with up to sub-millimeter resolution [87]. The NIRF imaging technique with the Indocyanine green (ICG) fluorophore has been widely used in the field of surgery, enabling the identification and localization of important tissues such as blood vessels, lymphatic vessels, and ureters [86,88]. Compared with white light, the NIRF imaging technique, with the addition of ICG fluorophores, allows for better optical contrast, higher signal/background ratio, and deeper photon penetration (up to 10 mm, depending on the tissue properties) [89,90,91].

The da Vinci surgical robot system (Intuitive Surgical, Sunnyvale, CA, USA) is equipped with a fluorescent imaging system called Firefly (Novadaq Technologies, Toronto, Canada), which allows surgeons to use NIRF imaging during robotic surgery to better visualize tissues such as blood vessels and extra-hepatic bile ducts [92]. Gioux et al. used a microscope equipped with an integrated NIRF system for lympho-venous anastomosis (LVA), and they found that the NIRF guidance during the microsurgery accelerated the surgeon’s identification and dissection of lymphatic vessels [93]. Using NIRF-based preoperative diagnosis and target marking, Mulken et al. successfully performed robotic microsurgical LVA using the MUSA microsurgery robot [66]. At present, there is no MSR system that uses intraoperative NIRF imaging to guide microsurgery.

Other imaging technologies

There are other imaging techniques that have the potential to be integrated with MSR systems to provide intraoperative guidance. One such technique is the probe-based confocal laser endomicroscopy (pCLE), which uses a fiber-optic probe to capture cellular level (up to 1 μm resolution [94]) images of in vivo tissues. It enables real-time, on-the-spot diagnosis and can generate larger maps of tissue morphology by performing mosaicking functions [95]. A handheld force-controlled robot was designed by Latt et al. to aid in the acquisition of consistent pCLE images during transanal endoscopic microsurgery [96]. Li et al. used pCLE as an end-effector for a collaborative robot, and a hybrid control framework was proposed to improve the image quality of robot-assisted intraocular non-contact pCLE scanning of the retina [97].

Another imaging technique that has gained popularity in recent years is the exoscope, which is essentially a high-definition video camera mounted on a long arm. It is used to observe and illuminate an object field on a patient from a location remote from the patient’s body, and project magnified high-resolution images onto a monitor to assist the surgeon [98]. In addition, the exoscope can be integrated with NIRF imaging techniques for improved surgical visualization [99]. Compared to the operation microscope, the exoscope has a longer working distance and provides better visual quality and greater comfort for the surgeon [100]. Currently, the effectiveness of the exoscope has been validated in neurosurgery and spinal surgery [98,101,102].

Ultrasound biomicroscopy (UBM) is another imaging technology that may be useful for MSRs. It utilizes high-frequency sound waves (35–100 MHz, higher than regular ultrasound) to visualize internal structures of the tissues with a high resolution (20 μm axial and 50 μm lateral for 50 MHz transducer), and the tissue penetration is approximately 4–5mm [103,104]. It is mainly used for imaging of the anterior segment of the eye [105]. Compared to OCT, which is also widely used in ophthalmology, UBM can provide better penetration through opaque or cloudy media, but it has relatively lower spatial resolution, requires contact with the eye, and is highly operator-dependent [106]. Integrating UBW technology into MSR systems can help surgeons to identify tissue features and provide intraoperative image guidance. In Table 2, we summarize the important parameters and the corresponding references of the imaging techniques mentioned above. Overall, the application of various imaging techniques to MSRs has the potential to increase the precision and accuracy of surgical procedures and improve surgical outcomes. Continued research and development in this area is likely to lead to even more advanced and effective MSR systems in the future.

#### 3.3.2. 3D Localization

Traditional microsurgery presents challenges for precise navigation and manipulation of surgical tools due to the limited workspace and top-down microscope view. One of the crucial obstacles is the 3D localization of the surgical instruments and intricate tissues such as the retina, cochlea, nerve fibers, and vascular networks deep inside the skull or spinal cord. Take ophthalmic microsurgery as an example; the lack of intraocular depth perception can significantly escalate the risk of penetration damage. To address this challenge, the adoption of high-precision target detection and advanced depth perception techniques becomes essential [5,10,108].

The 3D localization task in microsurgery can be divided into two aspects: target detection and depth perception. The target detection requires finding the target, such as the tool tip or blood vessels, from different types of images, while the depth perception involves analyzing the 3D position information of the target from the image. This section will provide a detailed introduction to both of these aspects.

Target detection

Numerous techniques emphasize target detection to accurately estimate an instrument’s position, thereby enhancing the clinician’s perceptual abilities. Initial attempts at automatic target detection depended on identifying instrument geometry amidst complex instrument appearance changes [109,110]. Recently, some methods leverage machine learning for rapid and robust target detecting and pose estimation [111,112,113,114]. Demonstrated autonomous image segmentation also offers the possibility of fully automated eye disease screening when combined with machine learning algorithms for ophthalmic OCT interpretation [115,116,117]. Deep learning-based methods have demonstrated strong performance and robustness in 2D instrument pose estimation [118,119]. Park et al. [120] suggested a deep learning algorithm for real-time OCT image segmentation and correction in vision-based robotic needle insertion systems, achieving a segmentation error of 3.6 μm. The algorithm has potential applications in retinal injection and corneal suturing.

An image registration-based pipeline using symmetric normalization registration method has been proposed to enhance existing image guidance technologies, which rapidly segments relevant temporal bone anatomy from cone-beam CT images without the need for large training data volumes [121]. In addition, several other image segmentation methods have been applied to retinal surgery. For instance, GMM [122] has been used for tool tip detection, while k-NN classifiers and Hessian filters [123,124,125,126], as well as the image projection network (IPN) [127] have been used for retinal vessel segmentation.

Depth perception based on microscope

Advanced target detection techniques in 3D space have substantially enhanced depth perception and procedural experience for the operating clinician [128,129]. For instance, Kim et al. leveraged deep learning and least squares for 3D distance prediction and optimal motion trajectories from a manually assigned 2D target position on the retina, demonstrating the effectiveness of deep learning in sensorimotor problems [130,131].

Some groups have used the relationship between the needle tip and its shadow to estimate the depth information [122,132]. Koyama et al. [133] implemented autonomous coordinated control of the light guide using dynamic regional virtual fixtures generated by vector field inequalities, so that the shadows of the instruments were always within the microscope view and the needle tip was automatically localized on the retina by detecting the instrument and its shadow at a predefined pixel distance. However, the accuracy of the positioning depends on the quality of the image, which affects the precise segmentation of the instrument and its shadows. Similarly, Richa et al. utilized the stereoscopic parallax between the surgical tool and the retinal surface for proximity detection to prevent retinal damage [134]. However, the effectiveness of this method is limited in fine ophthalmic surgery due to the rough correlation of 5.3 pixels/mm between parallax and depth. Yang et al. used a customized optical tracking system (ASAP) to provide the tool tip’s position and combined it with a structured light estimation method to reconstruct the retinal surface and calculate the distance from the tip to the surface, thereby achieving monocular hybrid visual servoing [135,136,137]. Bergeles et al. considered the unique optical properties and introduced a Raxel-based projection model to accurately locate the micro-robot in real-time [138,139]. However, the servo error is still several hundred microns, which is too large for microsurgical tasks.

Depth perception based on other imaging methods

Some researchers are exploring ways to improve the positioning accuracy of the tool tip by utilizing various imaging techniques. Table 3 briefly describes the main points of depth perception methods along with their corresponding references. Bruns et al. [140,141] proposed an image guidance system that integrates an optical tracking system with intraoperative CT scanning, enabling real-time accurate positioning of a cochlear implant insertion tool with a mean tool alignment accuracy of 0.31 mm. Sutherland et al. [4,37] proposes the NeuroArm system for stereotactic orientation and imaging within the magnet bore, with microsurgery performed outside the magnet. By integrating preoperative imaging data with intraoperatively acquired MRI scans, the robotic system achieves precise co-localization within the imaging space. After updating the images based on spatial orientation at the workstation with tool overlay, the surgical impact on both the lesion and the brain can be visualized. Clarke et al. proposed a 4-mm ultrasound transducer microarray for imaging and robotic guidance in retinal microsurgery [142], capable of resolving retinal structures as small as 2 μm from a distance of 100 μm. Compared to other imaging methods, the use of high-resolution OCT imaging information at the 3D location of the tool tip is potentially even more advantageous. Some groups enable the direct analysis of instrument–tissue interaction directly in OCT image space, eliminating the need for complex multimodal calibration required with traditional tracking methods [143,144].

In epiretinal surgery, Zhou et al. utilized microscope-integrated OCT to segment the geometric data model of the needle body, thereby facilitating marker-less online hand-eye calibration of the needle, with a mean calibration error of 9.2 μm [146]. To address the situation where the tip of the needle extends beyond the OCT scanning area, the same group proposed a spotlight projection model to localize the needle, enabling 3D instrument guidance for autonomous tasks in robot-assisted retinal surgery [147]. For subretinal surgery, the team used the reconstructed needle model over the retina to predict subretinal positioning when needle deformation was minimal [29]. For cases where the needle deformation could not be ignored, they proposed a deep learning-based method to detect and locate the subretinal position of the needle tip, and ultimately reconstruct the deformed needle tip model for subretinal injection [145].

#### 3.3.3. Force Sensing

Haptic is one of the important human senses. The temporal resolution of human touch is about 5 ms and the spatial resolution at the fingertips down to 0.5 mm [150]; thus, humans can acquire a wealth of information through the touch of the hand. However, haptic sensing is often lacking in most commercially available surgical robot systems (e.g., the da Vinci Surgical System). Force sensing technology can provide effective assistance in precise and flexible microsurgical operations, including determining tension during suture procedures [151], assessing tissue consistency in tumor resection surgeries [152], and executing the membrane peeling procedure in vitreoretinal surgery with appropriate force [153]. In current research, two primary types of force sensing technologies are employed in the MSR systems: those based on electrical strain gauge and those based on optical fibers.

Electrical strain gauge-based force sensors

The electrical strain gauge sensors measure force by detecting small changes in electrical resistance caused by the deformation of a material under stress, offering advantages such as wide measurement range and good stability. Some MSR systems connect the commercial electrical strain gauge sensors to their end-effector to enable force sensing during surgical procedures [53,152,154].

Take the NeuroArm robot system as an example; Sutherland et al. equipped each manipulator of the robot with two titanium 6-DOF Nano17 force/torque sensors (ATI Industrial Automation Inc., Apex, NC, USA), allowing for the measurement of tool–tissue interaction forces during neurosurgical procedures [152]. These force sensors were attached to the robotic tool holders, and each were capable of 0.149 g-force resolution, with a maximum threshold of 8 N and a torque overload of ±1.0 Nm. Due to the size limitation, this type of sensor is difficult to install near the tip of the surgical tool, which makes the measured data vulnerable to external force interference. Taking retinal surgery as an example, there is friction between the trocar and the tool, making it difficult for the externally mounted sensor to truly reflect the force between the instrument-tip and the target tissue [5].

Optical fiber-based force sensors

The optical fiber-based force sensors, on the other hand, measure the force by detecting the changes in light properties (wavelength and intensity) due to the external strain. Most of the fiber optic force sensors have the advantages of very slim size, high accuracy, biocompatibility, and sterilizability, and can be mounted distally on the surgical tools to provide more accurate force information [155]. In the field of optical fiber sensing, three categories of techniques can be identified based on their sensing principle: fiber Bragg grating (FBG) sensors, interferometer-based optical fiber sensors, and intensity-modulated optical fiber sensors [156]. Compared to other fiber optic force sensors, the FBG sensor exhibits higher precision, faster response time, and convenient multiplexing capabilities [156,157], which has led to its widespread adoption in MSR, especially in vitreoretinal surgery robots.

Bell et al. [158] integrated the FBG sensors into the neck of microforces to measure the 1-DOF crimp forces during stapedectomy procedures in otolaryngology, and they also integrated the force-sensitive microforces into their robot system to enable precise control of crimp force and forceps positioning. Iordachita et al. embedded FBG sensors into the shafts of surgical instruments to directly measure the contact forces between the instruments and ocular tissues in different DOF. They developed a series of force-sensing surgical instruments, including surgical forceps and hooks that detect lateral forces (2-DOF) [159], detect lateral and axial forces (3-DOF) [160,161], and detect contact forces between the tool tip and the retina as well as between the tool shaft and the sclera (4-DOF) [162]. With lateral resolution down to 0.15 mN, these force-sensing instruments can be used directly by surgeons or integrated into robotic systems to provide feedback for robotic motion [160,161,162,163]. To improve the effectiveness of retinal vein cannulation, researchers at JHU and CMU have incorporated FBG sensors into the MSR systems to identify the moment of venipuncture and maintain stability during the cannulation process [164]. Gijbels et al., from KU Leuven, have combined the FBG and OCT A-scan technologies and developed a novel cannulation needle, which can achieve the instrument–tissue interaction force perception (2-DOF, 0.2 mN resolution) as well as depth estimation during retinal vein cannulation [165,166].

### 3.4. Human–Machine Interaction (HMI)

#### 3.4.1. Force Feedback

The integration of advanced force sensing and feedback techniques into robotic surgery can help surgeons perceive the surgical environment and improve their motion accuracy, thereby improving the outcome of the procedure [155]. Force sensing techniques in MSR systems have been previously discussed, and this section will focus on the force feedback techniques that transmit the force information to the surgeon.

Compared to the widely used visual display technology, the application of force feedback technology in surgical robots is relatively immature [151]. Enayati et al. point out that that the bidirectional nature of haptic perception is a major difficulty preventing its widespread application [167]. The bidirectional nature of haptic perception means that there is a mechanical energy exchange between the environment and the sensory organ; thus, the inappropriate force feedback may interfere with the surgeon’s intended movement. But this also opens new possibilities for HMI. For example, the virtual fixture function can be achieved by combining the force feedback with different perception techniques (e.g., force or visual perception), which can reduce unnecessary contact forces or guide the surgeon’s movements [4,162]; it will be further described in the subsequent section.

The force feedback in MSR systems can be generally divided into two methods, direct force feedback and sensory substitution-based feedback.

Direct force feedback

For the direct force feedback method, the interaction forces between the tool and the tissue are proportionally fed back to the surgeon through haptic devices, so as to recreate the tactile sensations of the surgical procedure.

There are many MSR systems that use commercially available haptic devices to achieve direct force feedback. The force feedback in the second generation of the NeuroArm neurosurgical system is enabled by the commercially available Omega 7 haptic device (Force Dimension, Switzerland), which uses a parallel kinematic design and can provide a workspace of ∅ 160 × 110 mm and translational/grip force feedback of up to 12 N/8 N [168]. With the haptic device, the NeuroArm system can achieve force scaling, virtual fixture, and haptic warning functions [4,39]. The force scaling function can help the surgeon to clearly perceive small forces and recognize the consistency of the tissue during the operation; the virtual fixture function can guide the surgical tool or prevent it from entering the dangerous zone that is defined by the MRI information; and the haptic warning function will alert the surgeon by vibrating when force exceeds the threshold to avoid tissue damage.

Commonly used haptic devices also include Sigma-7 (Force Dimension, Nyon, Switzerland), HD2 (Quanser, Markham, ON, Canada), and PHANToM Premium 3.0 (Geomagic, Research Triangle Park, NC, USA). Zareiniaet al. conducted a comparative analysis of these systems and found that PHANToM Premium 3.0, which has a similar kinematic structure to the human arm, exhibited the best overall performance [169].

Meanwhile, some MSR systems employ custom-designed haptic devices to achieve the direct force feedback. Hoshyarmanesh et al. designed a microsurgery-specific haptic device [170], which features a 3-DOF active human arm-like articulated structure, a 4-DOF passive microsurgery-specific end-effector (3-DOF gimbal mechanism, 1-DOF exchangeable surgical toolset), and 3 supplementary DOF. The haptic device provides 0.92–1.46 mm positioning accuracy, 0.2 N force feedback resolution, and up to 15 N allowable force.

Gijbels et al. [3] developed a teleoperated robot for retinal surgery with a novel 4-DOF RCM structure slave arm and a 4-DOF spherical master arm. All DOFs of the master arm are active, so that the functions of active constraints and scaled force feedback can be implemented in the system to assist surgeons during retinal surgery.

Based on a cooperative MSR “Steady-hand Eye Robot (SHER)”, Balicki et al. [171] implemented the force scaling and velocity limiting functions. For the velocity limiting function, the threshold of velocity will decrease when the tool–tissue interaction force increases, so as to minimize the risk of retinal damage during the membrane peeling procedure. Uneri et al. [6] developed a new generation of SHER, which had a mechanical RCM structure and a larger workspace, and the robot system utilized real-time force information to gently guide the surgeon’s tool movements towards the direction of lower applied force on the tissue. On this basis, He et al. [162] further optimized the system and implemented an adaptive RCM constraint, thus reducing the force of the tool on the sclera during the surgery.

Sensory substitution-based force feedback

For the sensory substitution-based force feedback method, the force information is indirectly conveyed through other types of feedback, such as vibration, sound, or overlaid visual cues. Although this type of approach is relatively unintuitive compared to direct force feedback, it is stable and does not interfere with the surgeon’s movements [172].

Many researchers have applied force-to-auditory sensory substitution methods to robotic systems, where the system emits sound when the interaction force exceeds a threshold [6,70,173] or changes the sound accordingly to the magnitude of the interaction force [163,171,174]. Gonenc et al. found that incorporating force-to-auditory feedback can effectively reduce the maximum force during the membrane peeling [174].

Talasaz et al. use the bar indicator with height and color variations to visually display the force information during the robot suturing task, and the surgical outcomes between the direct force feedback and the vision-based force feedback are compared [175]. The results show that the vision-based force feedback can enhance the quality of suture knots and improve pulling force consistency, and the direct force feedback can reduce tissue stress and prevent accidental hits.

A wearable haptic device that can be worn on the finger pad and apply cutaneous force feedback is developed by Prattichizzo et al. [176], and it has been utilized to provide force information in surgical training tasks [177]. Aggravi et al. combined an Omega 6 haptic device (Force Dimension, Switzerland) and a wearable vibrotactile device to achieve the simultaneous direct force feedback and the sensory substitution-based feedback, and a haptic teleoperation system for flexible needles insertion was developed [178]. They compared different feedback types and found that the best performance was achieved by providing navigational information by kinesthetic feedback and tip cutting force by cutaneous vibrotactile feedback.

#### 3.4.2. Improved Control Performance

In microsurgery, the precision and stability of the surgeon’s control of the surgical instruments are critical to the outcome of the procedure. Riviere and Jensen observed that the RMS amplitude of tremor during manual instrument manipulation by surgeons is approximately 182 μm [23]. This level of tremor presents a significant challenge for the precise execution of microsurgery, which requires an accuracy of up to 25 μm [1,30]. Therefore, MSRs typically provide functions to enhance the control performance of surgeons on surgical instruments, including tremor filtering, motion scaling, and virtual fixtures.

Tremor filtering

Tremor filtering is a common function in most MSRs that helps to minimize unintended hand tremor of surgeons during surgical procedures. The basic principle of tremor filtering is to use various methods to filter out the high frequency signal caused by tremor in control commands. For teleoperated robots, the tremor filtering function can be achieved by applying a low-pass filter between the master module and the slave module. The NeuroArm robot, for example, processes the command signals through a low-pass filter, and surgeons can adjust its cut-off frequency according to their hand tremor for better tremor filtering [4]. For co-manipulated robots, the surgeon and the robot jointly control the movement of the surgical instruments, and the surgeon’s hand tremor can be effectively damped by the stiff structure of the robot arm [162]. Gijbels et al. designed a back-drivable co-manipulated robot, which allows the surgeon to control the manipulator without the need for a force sensor on the mechanism [179]. The tremor filtering function is achieved by adding virtual damping to the manipulator, and the surgeon can dynamically adjust the damping of the manipulator with a pedal for different surgical procedures. To balance control accuracy and flexibility, some researchers have proposed variable admittance-based control methods, where the system adjusts the admittance parameters according to the magnitude of the applied force [180] or the distance to the target [162] to achieve a better trade-off between compliance and accuracy. For handheld robots, the tremor filtering function is achieved primarily through a compensation algorithm, which drives the actuators between the tool tip and the surgeon’s hand to counteract unwanted tremors.

Motion scaling

Motion scaling is another function for control performance improvement, which scales down the surgeon’s hand movements by a certain level to relatively improve the precision. This function can be easily implemented into the teleoperated robots by processing the command signals before they are sent to the slave module [3,4,181]. Some handheld robots also have motion scaling functionality, but due to the small workspace of their manipulator, this feature can only be applied within a limited operational space [182]. Additionally, the co-manipulated robots do not support the motion scaling function due to the fixed position of the tool tip relative to the operator’s hand. The use of the motion scaling function can improve surgical accuracy, but it also reduces the range of motion of the surgical instrument, making it difficult to maneuver the instrument to reach distant targets during surgery. Zhang et al. proposed a motion scaling framework that combines preoperative knowledge with intraoperative information to dynamically adjust the scaling ratio, using a fuzzy Bayesian network to analyze intraoperative visual and kinematic information to achieve situation awareness [183]. The same group of researchers designed a teleoperated MSR with a hybrid control interface that allows the operator to switch between position mapping mode (with motion scaling) and velocity mapping mode via buttons on the master controller [49]. In velocity mapping mode, the MSR system controls the speed of the surgical instrument based on the force applied by the operator to the master controller, which facilitates its large-distance motion.

Virtual fixture

In addition to the above features to improve surgical precision, some MSR systems can provide customized constraints to improve the control performance and safety of robotic surgery through the virtual fixture function. The virtual fixtures rely on perceptual information, such as vision, force, position, and orientation, to create boundaries or guidance paths for surgical instruments in intricate scenarios, especially micromanipulation and fine manipulation in micron-sized workspaces. Some primary advantages of virtual fixtures include: the ability to easily define and modify fixture characteristics and dynamics within the software (such as stiffness, coarse, and viscous environment), the absence of mass or mechanical constraints, the elimination of maintenance needs, and the capability to easily develop, customize, and adapt these fixtures based on the surgical corridor for a specific patient [4,184].

Virtual fixtures, when designed around the operator’s objectives, can maximize surgical success and minimize incidental damage. For instance, in ophthalmology, virtual fixtures significantly reduce positioning error and the force applied to tissue in intraocular vein tracking and membrane peeling experiments [185]. Dewan et al. [186] used a calibrated stereo imaging system and surface reconstruction to create virtual fixtures for surface following, tool alignment, targeting, and insertion/extraction in an eye phantom. Similarly, Yang et al. [187] developed an eye model grounded in anatomical parameters, dividing the dynamic virtual constraint area to address cataracts caused by eyeball rotation and establishing unique force feedback algorithms for various surgical areas. This method does not require physical force sensors and meets the actual surgical requirements, reducing the complexity and cost of the surgical robot.

Furthermore, several groups have employed structured light and instrument tip tracking to maintain a constant confrontation distance [136,147]. Utilizing an OCT-guided system, Balicki et al. [188] precisely positioned the anatomical features and maintained a constant distance offset from the retina to prevent any potential collisions. More recently, the PRECEYES surgical system has incorporated similar fixed intraocular boundaries [189]. Kang et al. [190] have also introduced an OCT-guided surgical instrument for accurate subretinal injections, which features a dynamic depth positioning function that continually guides the syringe needle to the desired depth.

Moreover, in a case study using a NeuroArm robot for glioma resection [4], virtual fixtures were paired with augmented force feedback to reduce potential positional errors, with a haptic high-force warning system added to enhance operational safety and performance (tool–tissue interaction forces less than 1 N). For submillimeter accuracy in ear, nose, and throat (ENT) surgery, a cooperatively controlled robotic system with an image-guided technique was proposed to establish virtual constraints and enforce safety barriers, avoiding contact with critical anatomy and ensuring adaptability for a variety of neurotological procedures such as mastoidectomy, and translabyrinthine approach [191,192].

#### 3.4.3. Extended Reality (XR) Environment

In the field of HMI, the efficiency and effectiveness of user interactions with digital interfaces crucially shape the overall user experience. Extended Reality (XR), which encompasses virtual reality (VR), augmented reality (AR), and mixed reality (MR) [193,194], represents the frontier of HMI with its potential to radically transform how we interact with digital content. To achieve a high-quality immersive experience in the XR environment, it is crucial to ensure that users have precise, natural, and intuitive control over virtual objects, and that virtual objects realistically reflect the physical world [195,196,197]. There are three key challenges, namely high-fidelity feedback, real-time processing, and multimodal registration [198,199], that need to be addressed to minimize interaction errors and ensure that users can accurately manipulate virtual objects in a manner that mirrors the real world. This section will describe the 3D reconstruction techniques that provide high-fidelity surgical information for the XR environment, and present the current state of research on the XR environment for preoperative training and intraoperative guidance.

High-fidelity 3D reconstruction

As an integral part of visualization and perceptual interaction in XR, 3D reconstruction is increasingly used to create precise models of difficult-to-access objects or environments for preoperative planning [200,201,202], further enhancing the interaction precision and fidelity in the XR environment. However, its limitations such as time consumption, cost, and the effect of imaging conditions on model accuracy present challenges for microsurgery.

Combined with 3D reconstruction technologies, the visualization of anatomical structures using XR can significantly improve the accuracy and intuitive presentation in microsurgery. Stoyanov et al. [203] introduced motion-compensated visualization to ensure perceptual fidelity of AR and emphasized the need for real-time tissue deformation recovery and modeling, along with incorporating human perceptual factors in surgical displays. Probst et al. [129] implemented automatic 3D reconstruction and registration of the retina in ex vivo pig-eyes based on stereo camera calibration and robotic sensors. Zhou et al. segmented the geometric data model of the needle body and reconstructed the retinal model with a point cloud using microscope-integrated intraoperative OCT, though it lacked details like retinal layer segmentation and blood flow information [29,145,146].

Some researchers have reconstructed 3D retinal surgical models using optical coherence tomography angiography (OCTA), which provides more blood flow information, for retinal angiography or retinal layer segmentation. These models can be used to diagnose the health of the intraocular retinal vasculature based on the foveal avascular zone, retinal vessel density, and retinal thickness [204,205]. Li et al. [127] presented an image projection network that accomplishes 3D-to-2D image segmentation in OCTA images for retinal vessel segmentation and foveal avascular zone segmentation.

XR for preoperative training

The use of XR platforms for preoperative training has become increasingly popular in microsurgical procedures, with several platforms available, such as Eyesi (VRmagic Holding AG, Mannheim, Germany) [206,207], PhacoVision (Melerit Medical, Linköping, Sweden) [208,209], and MicrovisTouch (ImmersiveTouch Inc, Chicago, IL, USA) [210,211]. These platforms have proven effective in reducing the learning curve for surgical teams and saving time and costs in surgeries [203,204,205]. Additionally, these platforms are also widely used in clinical training, diagnosis, and treatment/therapy [212,213]. However, most of the platforms primarily utilize AR to incorporate comprehensive intraocular tissue information for preoperative training, but their use during intraoperative procedures remains limited.

In addition, some research teams have developed XR environments for preoperative training that accompany specific surgical robots. For instance, to improve efficiency and safety in preretinal membrane peeling simulation, Francone et al. [153] developed a haptic-enabled VR simulator using a surgical cockpit equipped with two multi-finger haptic devices. This simulator demonstrated reduced task completion time, improved tool tip path trajectory, decreased tool-retina collision force, and minimized retinal damage through haptic feedback. Sutherland et al. [214] have also developed a VR simulator for the NeuroArm system, and a study indicated that gamers adapted quickly to the simulator, potentially attributed to enhanced visual attention and spatial distribution skills acquired from video game play. Consequently, visuospatial performance has emerged as a crucial design criterion for microsurgical visualization [215].

XR for intraoperative guidance

The XR environment for intraoperative guidance requires timely feedback of surgical information to the surgeon, which places high demands on the efficiency of data processing and rendering in the MSR system. Many teams anticipate that combining intuitive and real-time presentation of intraocular tissue information using OCT in ophthalmic surgery will potentially address this issue. Seider et al. [216] used a 4D OCT system to generate a 3D surgical model with real-time surgical data, projected into the surgical telescope as B-scan and stereoscopic OCT volume for high-detail, near-real-time volumetric imaging during macular surgery. However, issues like high cost, limited scanning range, and challenging real-time performance of OCT remain to be tackled. Several teams are working on these challenges. Sommersperger et al. [217] utilized 4D OCT to achieve real-time estimation of tool and layer spacing, although such systems are not readily accessible

Furthermore, many research groups hope to integrate OCT imaging with XR for real-time intraoperative ophthalmic surgery in microsurgical workflows to compensate for intraocular information perception limitations. In AR-based surgical navigation for deep anterior lamellar keratoplasty (DALK), Pan et al. [218] introduced a deep learning framework for suturing guidance, which can track the excised corneal contour through semantic segmentation and reconstruct the missing motion caused by occlusion using an optical flow inpainting network. Draelos et al. [219,220] developed a VR-OCT viewer using volume ray casting and GPU optimization strategies based on texture trilinear interpolation, nearest neighbor interpolation, gradient packing, and voxel packing. It improves the efficiency of data processing without significantly degrading the rendering quality, thus bringing the benefits of real-time interaction and full-field display. Unlike traditional microscope systems which constrain the head of the surgeon to the eye-piece, an extended reality-based head-mounted device allows the surgeon to move their head freely while visualizing the imagery.

Another challenge in the application of XR for intraoperative guidance is the identification of essential information for augmentation in each surgical step, and the registration of multimodal images is not easy. To address this challenge, Tang et al. achieved 2D and 3D microscopic image registration of OCT using guiding laser points (fiducials) in AR [221], which eases intraocular surgery information perception for surgeons and improves surgery visualization. Roodaki et al. [144] combined AR and OCT to perform complex intraoperative image fusion and information extraction, obtaining real-time information about surgical instruments in the eye to guide surgery and enhance its visualization.

### 3.5. Automation

In the field of MSR, autonomous or semi-autonomous robots can not only respond to the surgeon’s commands, but also perform certain tasks automatically under the surgeon’s supervision, which can further improve surgical precision and efficiency and reduce fatigue caused by prolonged surgery [222]. The rise of automation in the surgical environment is increasingly being integrated into robotic microsurgery, particularly in ophthalmic surgery [223]. This section will provide a detailed presentation of the current automated methods and potential applications of automation in the MSR system.

Current automated methods

Current autonomous applications mainly address individual aspects of surgical procedures rather than covering the entire process [16,224]. Some MSR systems divide the process into subtasks, some of which are automated under the supervision, while others require manual intervention due to the limitations of existing technology. For example, cataract surgery encompasses six steps, including corneal incision, capsulorhexis, fragmentation, emulsifying and aspirating lens material, clearing the remaining lens material, and implant insertion [57]. The first three steps and implant insertion have been investigated for potential automation [225,226,227]. However, emulsifying lens removal and implant insertion, being delicate and hazardous, face challenges such as hand tremors and the absence of depth information. To address these challenges, Chen et al. [57] developed an MSR system called IRISS to achieve the semi-automated OCT-guided cataract removal. The system can generate anatomical reconstructions based on OCT images and generate tool trajectories, then automatically insert the instrument under image guidance and perform cataract removal tasks under the surgeon’s supervision. Subsequently, the same group utilized convolutional neural networks to accurately segment intraocular structures in OCT images, enabling successful semi-automatic detection and extraction of lens material [228]. In addition, polishing of the posterior capsule (PC, approximately 4–9 μm thick) reduces complications but is considered a high-risk procedure [229]. To address this, Gerber et al. developed the posterior capsule polishing function for IRISS, incorporating the guidance provided by OCT volume scan and B-scan [85].

Moreover, some research teams are focusing on automating preoperative operations to further improve precision and efficiency. A self-aligning scanner was proposed to achieve fully automatic ophthalmic imaging without mechanical stabilization in mannequins and free-standing human subjects, accelerating the automation process [220]. By combining laser-based marking, Wilson et al. [230] achieved the visualization of the otherwise invisible RCM to enable automatic RCM alignment with precision of 0.027 ± 0.002 mm, facilitating fully automated cataract extraction.

However, autonomous positioning and navigation pose challenges in microsurgery due to limited workspace and depth estimation. To overcome these difficulties, autonomous positioning of surgical instruments based on coordinated control of the light guide is being explored in RVC [133]. A spotlight-based 3D instrument guidance technique was also utilized for an autonomous vessel tracking task (an intermediate step in RVC) with better performance than manual execution and cooperative control in tracking vessels [147]. A technique that utilizes a laser aiming beam guided by position-based visual servoing, predicated on surface reconstruction and partitioned visual servoing, was employed to guide Micron for autonomous patterned laser photocoagulation—with fully automated motion and eye movement tracking—in ex vivo porcine eyes [136,231,232]. This method yielded results that were both quicker and more precise compared to semi-automated procedures and unassisted manual scanning of Micron. Additionally, image-guided MSR systems have been developed for automated cannulation of retinal vein phantoms, achieving precise targeting within 20 μm and successful drug infusion into the vascular lumen in all 30 experimental trials [22].

Potential automated methods

In the field of microsurgery, the integration of machine learning techniques, such as Learning from Demonstration (LfD) and Reinforcement Learning (RL), can enable robots to understand the surgical environment and perform appropriate operations, potentially advancing surgical automation and improving surgical outcomes [13]. LfD learns to perform tasks, and obtains policies from expert demonstrations [233]. The agent establishes a mapping relationship between state and action so that the agent can perform the appropriate action based on the state decision made at the moment. The expert is not required to provide additional data or any reinforcement signals during the training process. But it has limitations that include requiring a long training period and being highly dependent on the experience of surgeons [234]. On the other hand, RL enables the robot to learn actions by interacting with its environment and seeking to maximize cumulative rewards without expert demonstrations [235]. In this framework, the agent’s actions result in state transitions within the environment. These states are evaluated by an interpreter, which determines the corresponding reward that should be given to the agent. And then the interpreter transmits the rewards directly to the controller to maximize the total reward to control goal achievement [236]. However, RL’s training process is complex, and specifying the reward function can be challenging.

As a potential application in microsurgery, the integration of RL and LfD for application in microsurgery has been verified by Keller et al., using an industrial robot for OCT-guided corneal needle insertion [237]. Their algorithm obtained the desired results with minimal tissue deformation and was applied to multiple corneas. However, the semi-automated detection and extraction task’s success heavily depends on the quality of image segmentation of anatomical structures during DALK. Shin et al. [231] implemented semi-automatic extraction of lens fragments by semantic segmentation of four intraocular structures in OCT images using a convolutional neural network.

In addition, the RL can be used to realize fast motion planning and automated manipulation tasks [238,239]. And the automated surgical tool trajectory planning [240] and navigation tasks [131] were realized by robots using LfD. This trained model could achieve tens-of-microns accuracy while simplifying complex procedures and reducing the risk of tissue damage during surgical tool navigation. Simultaneously, Kim et al. [130] combined deep LfD with optimal control to automate the tool navigation task based on the Micron system, predicting target positions on the retinal surface from user-specified image pixel locations. This approach can estimate eye geometry and create safe trajectories within these boundaries.

## 4. Classic MSR Systems

In this section, several classic MSR systems will be introduced in detail from a more specific point of view. The main features and advancements in these systems will be presented to provide an in-depth discussion of the current state of research in the field of MSR.

### 4.1. NeuroArm System

The NeuroArm is an MRI-compatible teleoperated MSR system for neurosurgery, capable of performing micro-neurosurgery and stereotaxy in the MRI room and providing intraoperative image guidance. The system was developed in collaboration with researchers at the University of Calgary and the Canadian space engineering company Macdonald Dettwiler and Associates (MDA) [37]. Details of the development of the NeuroArm system can be found in Reference [241].

The structure of the NeuroArm robot is shown in Figure 5a, which includes two 7-DOF robotic arms that allow for rapid tool change, a field camera, and a cart on which the above devices are mounted. The robot was fabricated from non-magnetic materials, such as titanium and polyetheretherketone (PEEK), with the objective of preventing magnetic fields or gradients from impairing its performance and to ensure that the robot does not cause a significant degradation in image quality [37,65]. This design allows the system to align the position of the robotic arm with intraoperative MRI scans, facilitating microsurgery outside the magnet and stereotaxy within the magnet bore. In addition, the system is capable of precise motion with a spatial resolution of 50 μm at a maximum speed of 200 mm/s and a payload of up to 750 g [37].

For sensing technologies, the NeuroArm system not only utilizes visual data from the camera and the MRI scanner, but also measures force data in real-time using titanium 6-DOF Nano17 force/torque sensors (ATI Industrial Automation Inc., Apex, NC, USA) attached to the tool holder [152]. These sensors are capable of measuring force with an accuracy of 0.149 g and a maximum of 8 N. In terms of HMI, the NeuroArm system can recreate the visual, acoustic, and tactile information of the surgical procedure for the surgeon in a workstation outside the operating room, as shown in Figure 5b. The surgeon controls the robotic arm through a pair of the 7-DOF Omega 7 haptic devices (Force Dimension, Switzerland), while simultaneously accessing intraoperative magnetic resonance images and real-time high-resolution 3D images of the surgical site through a series of 2D and 3D displays [152,168]. The NeuroArm system incorporates various features to enhance the control performance of the surgical procedure. These include motion scaling, a 2 Hz low-pass filter for tremor filtering, and force scaling to increase tactile feedback sensitivity [4]. In addition, the system has implemented virtual fixture and haptic warning functions to provide intraoperative guidance based on tactile feedback. The former can define no-go zones based on MRI information, and the latter notifies the surgeon when interactive force exceeds certain thresholds.

Following preclinical evaluations [242] using rat and cadaver models for microsurgery and surgery within the bore of magnet, respectively, the NeuroArm system has been utilized in dozens of clinical cases of microsurgery, dealing with a range of pathologies such as glioma, meningioma, abscess, and radiation necrosis [39,152,243]. To further reduce system size, improve manipulation precision, and enhance HMI performance, the next generation of the NeuroArm is currently under development [241]. The research team has proposed a prototype haptic device specifically for microsurgery to improve ergonomics and reduce training limitations for inexperienced users [170].

### 4.2. REMS

The Robotic ENT Microsurgery System (REMS) is a co-manipulated MSR system designed for otolaryngology-head and neck surgery. The system was developed by the Laboratory for Computational Sensing and Robotics at Johns Hopkins University and is currently under a commercial license agreement with Galen Robotics Inc. (Baltimore, MD, USA) [244].

As shown in Figure 6, the REMS is a robotic platform that consists of a delta platform and three rotational joints; it can stabilize the operator’s primary instrument along 6-DOF and allows for rapid tool change [30,64,245]. The system incorporates a 6-DOF force/torque sensor (ATI Industrial Automation Inc., Apex, NC, USA) with force and torque ranges of 5 N and 500 mNm, respectively [246,247]. This integration enables the recording of force data and facilitates the system’s ability to provide warnings of excessive force and workspace limitations, which can promote safety during the operation. In addition, the system allows the surgeon to perform tremor-free movements by introducing resistance [248,249], and the robot’s velocity is controlled by scaling the gains from the force sensor using a foot pedal. With a resolution of 0.011 mm and a repeatability error of 0.302 mm for a 100 mm long tool, the system provides high precision and accuracy for surgical procedures. For more detailed information on the REMS platform, readers can refer to the available literature [250].

The feasibility and efficacy of REMS have been demonstrated in performing different tasks such as mastoidectomies, microvascular anastomosis, and stapedotomy [248,251]. In REMS-assisted mastoidectomies, the system can establish virtual fixation points with sub-millimeter accuracy by combining the intraoperative 3D Slicer with preoperative CT image [252]. This approach is designed to improve robot-assisted surgery by utilizing image-guided and preoperative planning systems to visualize relevant anatomical structures comprehensively [253]. The REMS platform has been extensively studied in human cadaver models, focusing on intraoperative sinus navigation using 3D Slicer in conjunction with preoperative reconstruction and planning [247,250]. The study yielded an average registration error of 0.46 mm with a standard deviation of 0.22 mm.

### 4.3. MUSA System

The MicroSure’s MUSA was developed by Maastricht University Medical Center in collaboration with Eindhoven University of Technology, which is the world’s first dedicated robotic platform for (super)microsurgery [254]. The MUSA is designed to aid in stabilizing movements of the microsurgeon by filtering tremors and scaling down motions. The robot is easily maneuverable, equipped with arms holding genuine microsurgical instruments that are easily placed into the holders, and are compatible with conventional surgical microscopes [66].

In reality, MUSA is a second-generation design for microsurgery. The MicroSure robot, as the first generation, was designed in 2014 by the same team. The prototype of MicroSure robot is composed of subunits [41]. The robotic arms are attached to the suspension ring and can be combined with instruments. The suspension ring can be attached to the operation table while the surgeon controls the master manipulators. MUSA is based on the design of the MicroSure robot, which is designed more ergonomically to minimize weariness and further improve manual dexterity. The components of MUSA are shown in Figure 7. The slave arm of the system has a total of seven degrees of freedom, with six degrees of freedom for positioning and orientation and one degree of freedom for actuating microsurgical instruments. The MUSA is a lightweight, small-sized system providing easy integration in the operating room with minimal adaptations of the room layout and organizational planning [254].

The system, as a master-slave platform, is based on a modular design, permitting it to be used in different configurations depending on the level of required robot assistance, which enables a quick switch between conventional manual mode and robot-assisted mode. It is able to enhance the surgeon’s physical performance by offering motion scaling and tremor filtration with the precision at the slave end effector of about 70 µm. Through optimization of the control software, a bidirectional precision down to 30–40 µm can be achieved. And there is an ability to change motion scaling settings in real-time [41].

In 2018, experiments demonstrated that it is feasible to complete anastomotic microsurgery on silicone vessels using the prototype of the MicroSure robot. The robot-assisted performance showed steeper learning curves [51]. In addition, an animal study in the same year confirmed the feasibility of performing a microvascular anastomosis on the abdominal aorta and femoral arteries in rats using the MicroSure robot, and the results are consistent with the previous results [255]. More recently, a prospective randomized pilot study was conducted comparing MUSA robot-assisted and manual supermicrosurgical lymphatico-venous anastomosis (LVA) in treating breast cancer-related lymphedema. The data provide the first-in-human proof that robot-assisted supermicrosurgical LVA in patients is in fact feasible and safe using the MUSA [66]. In the future, the research objectives are the incorporation of enhanced tactile feedback in the system to enhance precision and the incorporation of machine learning in the system to endow robotic systems with the ability of self-learning and cognitive skills, thereby enabling automation [254].

### 4.4. IRISS

The intraocular robotic interventional surgical system (IRISS) is a teleoperated intraocular MSR system developed by a team from the University of California, Los Angeles (UCLA). The long-term goal of the project is to develop a surgical robot system that combines augmented reality teleoperation and full automation to perform anterior and posterior intraocular surgical procedures [256].

As shown in Figure 8, the slave manipulator of the IRISS consists of two 7-DOF independently controllable robotic arms that can be controlled independently, each with a mechanical RCM structure based on curved guides [257]. The robotic arms are driven by brushed DC motors and are capable of achieving a positional precision and accuracy of about 27 ± 3 μm and 205 ± 2 μm, respectively [233]. Additionally, each robotic arm of the IRISS can hold two surgical instruments that can be automatically exchanged outside the eye.

For HMI, the surgeon remotely operates the slave manipulator through a pair of customized master controllers, and observes the intraoperative 3D visual feedback through a heads-up monitor, and the visual information is obtained through a surgical microscope mounted with a stereo camera (TrueZoom 3D, TrueVision, Indianapolis, IN, USA) or an SD-OCT imaging system (Telesto II 1060LR, Thorlabs, Newton, NJ, USA) [55,257]. In addition, the system provides tremor filtering and motion scaling functions to enhance the control performance and safety of robotic surgery. The performance of IRISS was validated in ex vivo porcine eyes, where surgeons operated IRISS to successfully perform a series of vitreoretinal procedures, including anterior lens capsulorhexis, vitrectomy, retinal vein cannulation, and other sophisticated procedures [233].

In recent research, investigators have applied OCT imaging technology to the IRISS system and developed a series of OCT and camera image-guided semi-automation/automation schemes for intraocular surgery. These include semi-automated lens removal [55,258], posterior capsular polishing [85], identification and removal of residual lens fragments [228], and automated retinal vein cannulation [22]; these schemes have been validated using ex vivo porcine eyes or silicone phantoms.

### 4.5. Preceyes Surgical System

The Preceyes Surgical System (PSS), developed by the Eindhoven University of Technology in the Netherlands [53,55] and later by Preceyes B.V., is the only commercially available robotic intraocular surgical system with CE mark certification [259]. The PSS is a teleoperated MSR system consisting of a computer, an input motion controller, an instrument manipulator, and a surgical table-mounted headrest (as shown in Figure 9). Designed for optimal performance, the system features a parallelogram linkage with an adjustable counterweight, providing a mechanical RCM, power outage protection, and minimized joint torques [260]. The PSS employs dynamic scaling to transform coarse movements into precise four-axis movements of the instrument tip. Furthermore, it utilizes distance boundaries based on OCT to prevent undesired motion and incorporates tremor filtering to reduce iatrogenic retinal trauma [261]. Additional features include haptic feedback, automated instrument changing, auditory feedback in proximity to the retina, and an augmented retraction mechanism for immediate probe removal in case of accidents [189]. These features improve precision, safety, and reduce the risk of inadvertent tissue damage [262].

The PSS achieves an intrinsic precision of 10 µm for the instrument tip and enables cannulation of approximately 60 µm diameter pig retinal venules [265,266]. It allows surgeons to use standard microscopes, such as M822 F40 Leica (Diegem, Belgium) [260], Zeiss Rescan 700 (Oberkochen, Germany) [262], or other visualization techniques like intraoperative OCT (Zeiss Resight 7000, Oberkochen, Germany). This integration offers valuable real-time imaging and enhanced visualization feedback to the surgeon during the procedure [263,265]. Moreover, when combined with the Eyesi surgical simulator, the PSS facilitates surgeons’ quick acquisition of basic competency, enabling skill development and assessment [189,213,267,268].

The world’s first robotic ILM peeling using the PSS was performed in Oxford in 2016, and additional clinical studies are underway [269]. Some studies involve robot-assisted or manual surgery in three patients requiring subretinal injections and 12 patients requiring dissection of the epiretinal or inner limiting membrane over the macula [265]. Additionally, the first-in-human robot-assisted subretinal drug delivery was performed on 12 patients, with six undergoing robot-assisted and six undergoing conventional manual techniques [262]. The PSS is also being evaluated in the world’s first controlled trial on robotic surgery for epiretinal membrane [270].

Compared to manual surgery, the PSS offers enhanced precision, although it operates at a slower pace [48,268,270]. Extensive preclinical evaluations have demonstrated the PSS’s ability to achieve ultrafine cannulation of retinal blood vessels and accurate injection of antithrombotic agents in live pigs [260,261,266]. It has also been successfully employed in clinical settings for patients with macular pucker, macular hole, and subretinal hemorrhage [262,265,270].

### 4.6. Co-Manipulator System

In 2014, the Micro- and Precision Engineering Group at the Catholic University of Leuven presented a teleoperated robotic system (Figure 10) complete with motion scaling, tremor compensation, and scaled force feedback [179]. It offers a precision enhancement of more than tenfold eye stabilization, and hand-free tool immobilization.

This system features a new Remote-Center-of-Motion mechanism with four degrees of freedom controlled through a spherical mechanism. Its slave manipulator consists of two subsystems: the surgical system and the alignment system. The surgical system, to which the surgical instrument is fixed, assists the surgeon in performing the treatment, keeping the eye from rotating during the intervention. The alignment system is used to pre-operatively align the RCM of the surgical system with the incision. Its master manipulator is filament-driven, and the isomorphic configuration reproduces the motion of the instrument in the eye [3].

The system provides two control modes, co-manipulation and tele-manipulation, both of which significantly enhance positioning precision compared to free-hand operation. In terms of user experience, the co-manipulation mode is generally preferred because of its improved precision and ease of use, although the telemanipulation mode demonstrates superior positioning precision [271].

Recent research has shown that the co-manipulated robotic system can improve surgical precision by generating motion-opposing forces that increase with the speed of motion. This strategy is typically implemented with the aid of a non-backdrivable device that is controlled using a handle-mounted force-sensor in an admittance scheme. However, the co-manipulator robotic system uses a backdrivable architecture, eliminating the need for the force sensor. This allows the surgeon to manipulate the system directly, with reaction forces generated based on the current tool speed [7]. In the detection of target eye tissue, image recognition and reconstruction are the main research methods. Researchers have also used image methods for the reconstruction of retinal vessels in the fundus.

This system was used to perform in vivo retinal vein cannulation on pig eyes in 2017 [22]. Complete success was defined as a stable intravenous position of the needle tip for more than 3 min, and was confirmed in 15 out of 18 eyes. Following this initial success, the same group reported on the clinical evaluation of their system in the world’s first-in-human, robot-assisted retinal vein cannulation [7]. In this study, four RVO patients have been treated with the technology in the context of a phase I clinical trial, and the results demonstrate the technical feasibility of the system in safely injecting an anticoagulant into a 100 μm-thick retinal vein for a period of 10 min.

### 4.7. Main Parameters of Classic MSR Systems

Previous sections have provided detailed introductions to several specific MSR systems. For more comprehensive information, Table 4 has been compiled to summarizes the main parameters of classic MSR systems, including their application scenarios, structural information, control methods, main functions, and research progress. Through this table, readers can better understand and compare the features and parameters of typical MSR systems in different specialties.

## 5. Current Challenges and Future Directions

### 5.1. Current Challenges

The technologies of the MSR have long been a matter of great interest in a wide range of fields, but there are still many difficulties that limit the widespread use of MSR systems. The main challenges faced by many researchers are summarized in Table 5.

As shown in the table, an existing difficulty for the introduction of MSR systems remains the inertia of ongoing medical practice [288], which requires particular effort to overcome. There is still a lot of work to be done to gain the trust of clinicians and patients. For the device to be marketable, it must also be accepted by third-party payers in the health-care system, including insurance companies [289]. Ethical and legal concerns arise when considering the clinical use of MSR [290]. One issue is the determination of responsibility in cases of medical malpractice. Compared with surgeons, surgical robots have no self-awareness during surgery, and if a malfunction occurs during surgery and damages the patient’s life and health, it will be very difficult to determine the responsibility for the medical malpractice. It is generally believed that the damage caused by the quality of the robot itself should be the responsibility of the manufacturer, and the doctor should be mainly responsible for the improper operation during the surgery. However, for such a new thing as MSR, internationally recognized regulatory standards have not yet been established. Another ethical concern is the principle of fairness in the allocation of medical resources. Given the existing medical conditions and level of economic development, hospitals that are in a position to use surgical robots are economically advanced and densely populated developed cities. Therefore, fair and effective allocation of medical resources is crucial. The development of surgical robots may intensify the disparity in the allocation of medical resources. There is a need for concerted social effort to maintain equal distribution of resources such that everyone receives equally high-quality medical care.

Regarding the development of MSR systems, it is highlighted that it requires an interdisciplinary approach. Throughout the development process, it is important for professional engineers to collaborate closely with surgeons, who are the actual end-users of the technology, to solve complex technical challenges while meeting real clinical needs. The success of robotic surgery relies on the effective interaction between the surgeon and the robot system. However, there is a significant trade-off associated with MSR systems, which is the potential reduction in direct surgeon –patient interaction, which may increase the risk of error. In this respect, human factors emerge as a most critical component to ensure safer, more usable, and effective devices, which must be an integral part of the design. Of relevance is also the development of user-interfaces that provide adequate information and effective control. A key enhancement is the use of haptics, allowing the operating physician to sense the forces applied by the manipulation system [153,282].

Microscope-based visual feedback can only provide a limited field of view, which can affect the surgeon’s operational accuracy and perception of the surgical environment, posing a challenge to the visualization of MSR systems. Recent advances allow the fusing of intraoperative images with patient-specific 3D models constructed using preoperative information for improved visualization [291]. In addition, merging imaging information acquired from different imaging modalities is another possibility that can be further exploited [55]. Control robustness is also an important aspect in the design of MSR systems, which should be able to handle unforeseen events or disturbances during surgery, such as unexpected target tissue motion, so as to ensure patient safety and the success of the procedure. The teleoperated type of MSR is highly dependent on network communication. The speed and stability of the network directly affect the quality of the procedures. Furthermore, the network security greatly affects the application of MSR in clinical practice, including the protection of patient privacy, as in the case of many other medical procedures facilitated by digital technology.

The wide use of MSR depends not only on the availability of the technology, but also on the availability of trained physicians. MSR requires specialized skills compared to traditional methods, and it is essential that medical schools be equipped with such technologies to train physicians appropriately. Addressing the above challenges is imperative for the widespread application of MSR systems and should be emphasized in prospective system development. Moreover, it is important to acknowledge the trade-offs inherent in MSR systems. Medical institutions and users need to balance the benefits and existing drawbacks of MSR systems. These drawbacks encompass high acquisition and maintenance costs, limited range of procedures that can be adapted, additional training requirements, and the reduced direct surgeon–patient interaction. Understanding and addressing these trade-offs within MSR systems will contribute to their efficient integration and use in the field of microsurgery.

### 5.2. Future Directions

Based on the literature survey and analysis, we propose three directions for the future development of MSR, which include further consideration of the human factors, multiple sensor fusion, and a higher level of autonomy.

#### 5.2.1. Further Human Factors Consideration

Future MSR research should prioritize human factors engineering, so as to improve the adaptability of robotic systems in the operating room and to meet surgeons’ needs. This includes focusing on the design of compact and highly compatible robotic systems, as well as intuitive HMI strategies. In terms of structural design, researchers need to take into account the practical surgical requirements, the surgeon’s pose, operating room layout, and limitations imposed by imaging devices. A compact robot structure helps reduce the overall device weight and minimizes the occupation of valuable surgical space [249,292]. This enhances the maneuverability of surgical robots and avoids obstructing the imaging devices such as the microscopes [241]. These factors are important for MSR systems that require precise operations in confined surgical environments. However, it is noteworthy that many research teams are currently developing MSR systems in laboratory settings with limited exposure to real surgical environments. This lack of direct exposure to actual surgical conditions poses a challenge when integrating these laboratory-developed MSR systems into existing clinical applications. Research teams should work closely with clinical surgeons to fully consider human factors and develop robotic systems that can be seamlessly integrated into real-world surgical environments.

In addition, there is room for further improvement in the HMI strategies for MSR. One promising direction is the application of extended reality (XR) interfaces to MSR systems, which have the potential to be utilized for preoperative surgical training as well as intraoperative surgical visualization [221,293]. Compared to observing the surgical environment through a microscope, integrating XR-based interfaces can assist surgeons in intuitively visualizing the surgical environment and operating the MSR system in an ergonomic manner. In addition, virtual fixture functionality can be enhanced, including the incorporation of XR interfaces to provide multi-sensory feedback (like haptic and visual cues) to provide improved intraoperative guidance for surgeons [4].

#### 5.2.2. Multiple Sensor Fusion

The fusion of multiple sensors is another area of research that could be further explored. By integrating information from different sensors during surgery, a more comprehensive and accurate depiction of the surgical scene can be achieved. For example, sensors such as microscopes, OCT, and force sensors that acquire different information of a target can be fused. This enables information-rich perception of the surgical field, helping surgeons or robotic systems to make informed intraoperative decisions. In addition, by integrating the intraoperative data with preoperative patient data (such as OCT, CT, and MRI data), surgeons can gain a more complete view of the surgical environment and achieve more accurate navigation [294]. This integration has the potential to provide significant benefits in complex surgical tasks. Therefore, researchers should focus on the development and integration of different types of sensors to enhance the perception capabilities of robotic systems. Additionally, it is important to develop more sophisticated fusion algorithms to efficiently process and dynamically update information from multiple sources in real-time.

#### 5.2.3. Higher Level of Autonomy

Elevating the level of autonomy is another developing trend in MSR. Currently, most MSR systems rely on manual manipulation by surgeons, which requires a high degree of surgical precision and hand-eye coordination. By increasing the autonomy level of the MSR system in the pre- and intraoperative period, the surgeon’s workload can be reduced, leading to improved surgical efficiency. For example, incorporating machine learning techniques, MSR systems can identify and analyze patient status preoperatively, thus enabling automatic intraoperative targeting or providing patient-specific surgical recommendations to the surgeon [55]. In the intraoperative phase, the application of automation technology allows the MSR to automatically perform some of the surgical tasks based on a predefined surgical path or adjust the path in real-time based on sensor feedback, leading to improved surgical efficiency [85].

For a higher level of autonomy, the focus is on leveraging the advancement in related fields such as machine learning and computer vision technologies to realize more sophisticated robot perception and autonomous control algorithms for MSR systems. In addition, to ensure the safety and reliability of the automated functions in practice, researchers need to develop appropriate monitoring and evaluation schemes that allow for human intervention when necessary to ensure safety in surgical procedures.

## 6. Conclusions

The utilization of MSR systems in microsurgery can greatly enhance surgical capabilities, providing surgeons with improved precision, stability, and controllability. These advancements can ultimately lead to more precise and safer surgeries.

In order to gain a comprehensive understanding of the application scenarios and practical requirements of MSR, our initial investigation focuses on the usage and challenges of microsurgery across various specialties. Through our research, we discovered that manual microsurgery faces four primary challenges: manipulation of small and delicate objects, limited perceptual feedback, non-ergonomic operating positions, and the necessity for extensive training.

Subsequently, we delved into MSR systems designed to address these issues, providing an overview of the key technologies in four aspects: the operation modes and mechanism designs which constitute the foundation of the robot system; the sensing and perception technologies that assist MSR in collecting environmental data; the human–machine interaction methods to improve the surgeon’s control performance and provide intuitive feedback; and the automation technologies that enhance surgical efficiency by enabling MSR to perform surgical tasks automatically or semi-automatically. In addition, the development, characteristics, and primary parameters of various classic MSR systems are presented from a more specific point of view.

Finally, we discussed the current challenges in implementing MSR systems and looked into three potential directions for prospective development in this field, including further human factors consideration, multiple sensor fusion, and a higher level of autonomy, where we provided insights on the potential role of artificial intelligence in advancing the level of autonomy in MSR. We envision that this review will provide readers with a thorough understanding of MSR-related technologies and thus inspire further innovation and research in the MSR field.

## Figures and Tables

**Figure 1 sensors-23-08503-f001:**
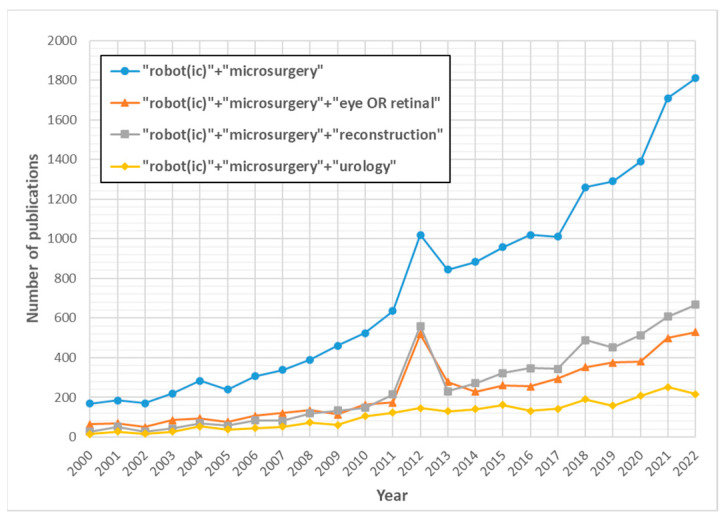
Number of MSR-related publications each year from 2000 to 2022 based on Google Scholar keyword search.

**Figure 2 sensors-23-08503-f002:**
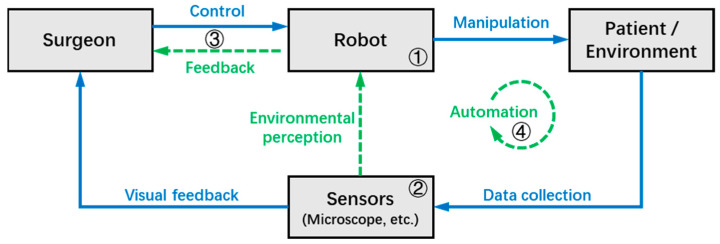
The typical workflow of the MSR system.

**Figure 3 sensors-23-08503-f003:**
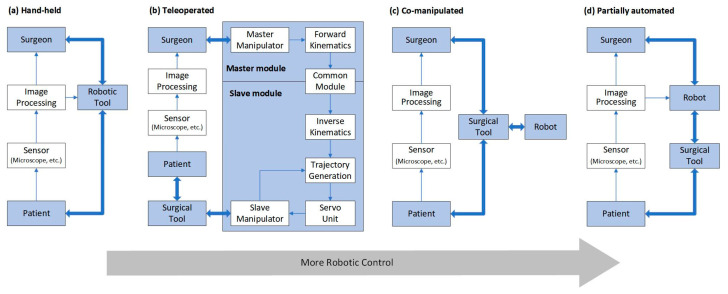
Types of surgical robots. (**a**) Handheld robot, (**b**) Teleoperated robot, (**c**) Co-manipulated robot, (**d**) Partially automated robot.

**Figure 4 sensors-23-08503-f004:**
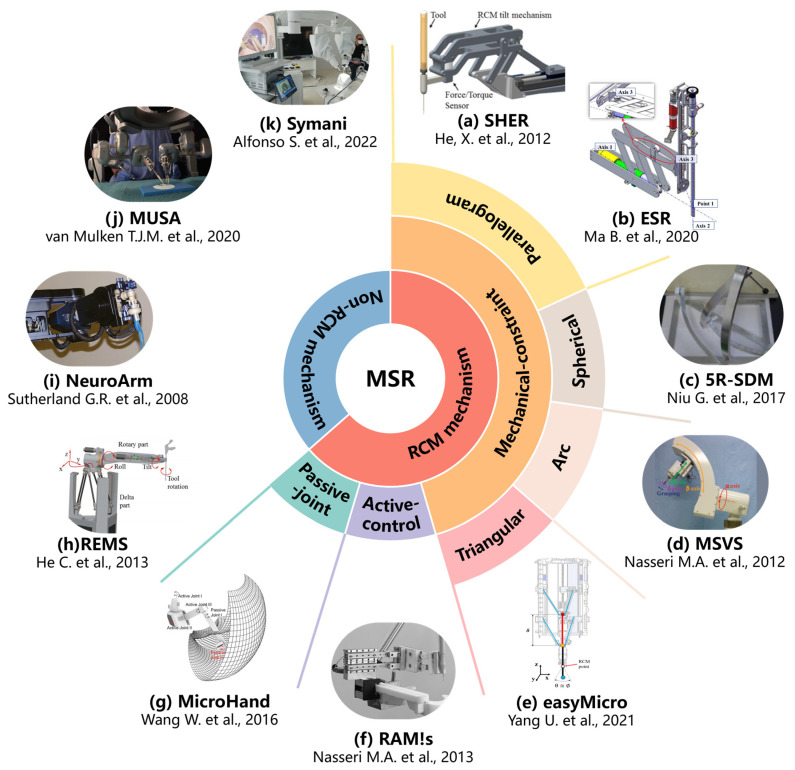
Typical structures of MSR. (**a**) SHER. Adapted with permission from Ref. [57]. Copyright 2012, ASME. (**b**) ESR [58]. (**c**) 5R-SDM. Adapted with permission from Ref. [59]. Copyright 2017, Elsevier. (**d**) MSVS. Adapted with permission from Ref. [60]. Copyright 2012, Springer Nature. (**e**) easyMicro. Adapted with permission from Ref. [61]. Copyright 2021, John Wiley and Sons. (**f**) RAM!s. Adapted with permission from Ref. [62]. Copyright 2013, IEEE. (**g**) MicroHand. Adapted with permission from Ref. [63]. Copyright 2016, John Wiley and Sons. (**h**) REMS. Adapted with permission from Ref. [64]. Copyright 2013, IEEE. (**i**) NeuroArm. Adapted with permission from Ref. [65]. Copyright 2008, IEEE. (**j**) MUSA [66]. (**k**) Symani [67].

**Figure 5 sensors-23-08503-f005:**
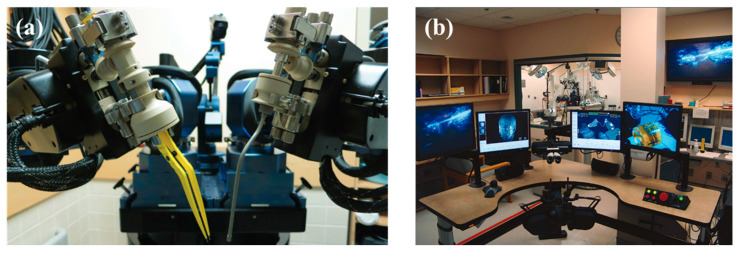
The NeuroArm system: (**a**) the robotic arms [4], (**b**) the NeuroArm workstation [65]. Adapted with permission from Ref. [65]. Copyright 2008, IEEE.

**Figure 6 sensors-23-08503-f006:**
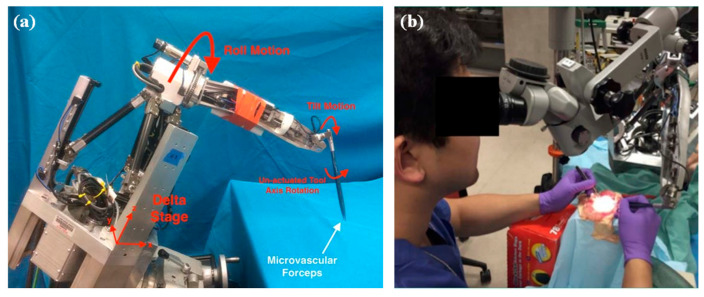
The REMS system: (**a**) the system structure, (**b**) the experimental setup. Adapted with permission from Ref. [248]. Copyright 2017, The American Laryngological, Rhinological and Otological Society, Inc.

**Figure 7 sensors-23-08503-f007:**
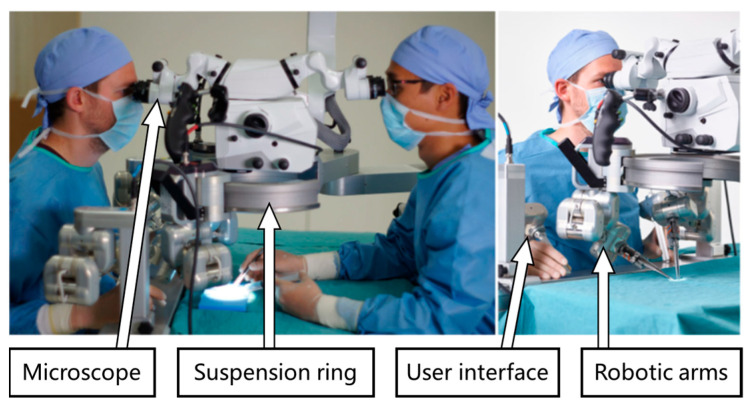
The Microsure’s MUSA robot [254].

**Figure 8 sensors-23-08503-f008:**
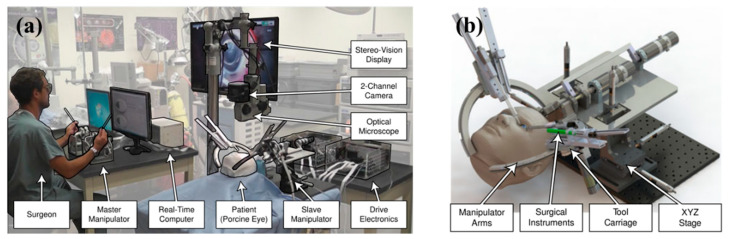
The intraocular robotic interventional surgical system (IRISS): (**a**) system setup, (**b**) CAD model of the slave manipulator. Adapted with permission from Ref. [233]. Copyright 2018, John Wiley and Sons.

**Figure 9 sensors-23-08503-f009:**
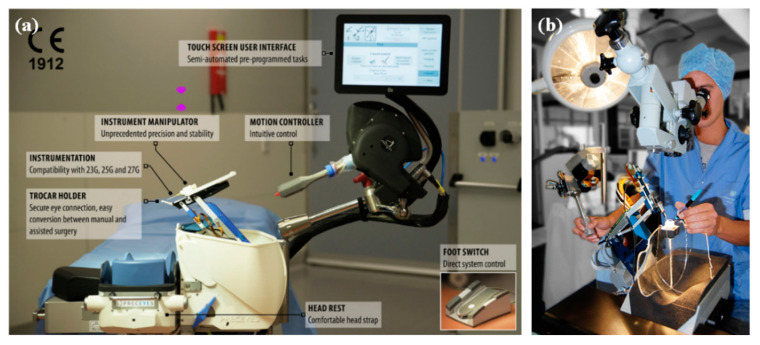
The Preceyes surgical system: (**a**) system setup. Adapted with permission from Ref. [263]. Copyright 2021, American Academy of Ophthalmology. (**b**) the experimental setup. Adapted with permission from Ref. [264]. Copyright 2013, Association for Research in Vision and Ophthalmology.

**Figure 10 sensors-23-08503-f010:**
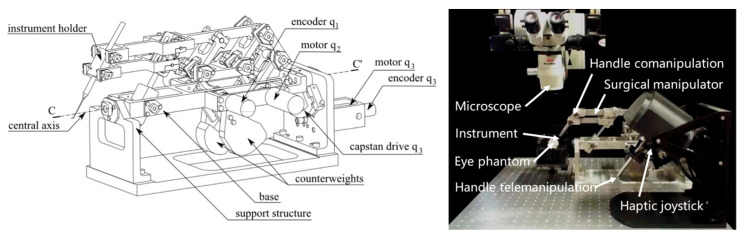
The co-manipulator robotic system. Adapted with permission from Ref. [3]. Copyright 2014, IEEE.

**Table 1 sensors-23-08503-t001:** Examples of clinical specialties requiring microsurgery.

Clinical Specialty	Example Procedure	Handling Target	Imaging Method	Challenges
Ophthalmology	Epiretinal membrane peeling [18]	Epiretinal membrane (thickness of 61 ± 28 μm)	Microscope, OCT	The small thickness of the membrane High-precision peeling
	Retinal vein cannulation [24]	Retinal vascular branch (∅ < 200 μm)	Microscope, OCT	The small tool–tissue interaction forces and fragility of retinal veins
	Subretinal injections [28]	Retina (foveal thickness < 250 μm)	Microscope, OCT	Lack of depth information High-precision injection
Otolaryngology	Cochlear implant surgery [31]	Cochlea implant	Microscope, CT	Small insertion force (<0.1 N) High cochlear placement accuracy requirement (≈0.5 mm)
	Endonasal skull base surgery [32]	Skull base lesion or tumor	Microscope, Endoscopes	Poor sensory feedback Narrow surgical passages Collisions between instruments
Neurosurgery	Surgical interventions for glioma [39]	Brain tumor	Microscope, MRI	Delicate brain tissue Difficult to distinguish tumor from normal tissue Prolonged uncomfortable positions
	Superficial temporal artery to middle cerebral artery (STA–MCA) anastomosis [40]	Blood vessel (∅ < 1 mm)	Microscope	High-precision anastomosis
Reconstructive Surgery	Deep inferior epigastric perforator (DIEP) [41]	Blood vessel (∅ 0.8–2.5 mm), nerve	Microscope	High-precision anastomosis
	Lympho-venous anastomosis (LVA) [44]	Vein and lymphatic vessel (∅ 0.3–0.8 mm)	Microscope, NIR	High-precision anastomosis Difficult to identify the lymphatic vessel
Urology	Vasoepididymostomy (VE) [47]	Vas deferens (lumen of 0.3–0.4 mm), epididymal tubule (∅ 0.15–0.25 mm)	Microscope, endoscope	High-precision anastomosis Need long period of training

**Table 2 sensors-23-08503-t002:** MSR-related imaging technologies.

Modality	Source	Resolution	Depth of Penetration	Main Clinical Application	References
Magnetic resonance imaging (MRI)	Magnetic fields	~1 mm	unlimited	Neurosurgery, Plastic surgery, etc.	[71,77]
Computed tomography (CT)	X-rays	0.5–1 mm	unlimited	Orthopedic, Neurosurgery, etc.	[72,107]
Optical coherence tomography (OCT)	Low-coherence light	5–20 μm	<3 mm	Ophthalmology	[79,107]
Near-infrared fluorescence (NIRF)	Near-infrared light	Up to sub-millimeter	<10 mm	Plastic surgery	[87,89,90,91]
Exoscope	Light	Determined by camera parameters	Determined by the imaging modality	Neurosurgery, Spinal surgery	[98,99,100,101,102]
Ultrasound biomicroscopy (UBM)	Sound waves	20–50 μm (For 50 MHz transducer)	<5 mm	Ophthalmology	[103,104]

**Table 3 sensors-23-08503-t003:** Different depth information perception methods.

Procedure	Method	Precision	Note	References
Subretinal surgery	Combining geometric information of the needle tip and OCT	4.7 μm (average distance error)	The needle has less deformation	[29]
	OCT, image segmentation, and reconstruction	99.2% (confidence of localizing the needle)	The needle has large deformation	[145]
Epiretinal surgery	Estimation of calibration parameters for OCT cameras	9.2 μm (mean calibration error)	Unmarked hand-eye calibration and needle body segmentation	[146]
	Spotlight-based 3D instrument guidance	0.013 mm (average tracking error)	For positioning the needle tip when it is beyond the OCT scanning area	[147]
	Monocular microscope, structured light, and customized ASAP	69 ± 36 μm	Hybrid vision and position control	[135,136,137]
	Stereomicroscope and 3D reconstruction	150 μm (translational errors of the tool)	Hand-eye calibration with markers and 3D reconstruction of the retina	[129,148]
	Stereomicroscopy and parallax-based depth estimation	5.3 pixels/mm	Inability to perform fine intraocular manipulation	[134]
	Depth perception based on deep learning	137 μm (accuracy of physical experiment)	Predict waypoint to goal in 3D given 2D starting point	[130,131]
	Automatic localization based on tool shadows	0.34 ± 0.06 mm (average height for autonomous positioning)	Dependent on image quality	[122,132,133]
	Raxel-based projection model	314 ± 194 μm	Positioning dependent on unique optical properties	[138,139]
Cochlear implant	Optical tracker and stereo cameras	0.31 mm (mean tool alignment accuracy)	Image-guidance paired with an optical tracking system	[140,141,149]
Glioma resection	Pre- and intraoperative image alignment	/	Stereotactic orientation and imaging within the magnet bore and microsurgery outside the magnet	[4,37]

**Table 4 sensors-23-08503-t004:** Main parameters of classic MSR systems.

System Name	Organization	Application	Mechanical Structure	DOF	Workspace		Precision (μm)	Size (mm)	Weight (kg)	Control	Function					Research Progress
					XYZ (mm)	Angular (◦)					Motion Scaling	Tremor Filtering	Force Feedback	3D Perception	Auto-mation	
SMOS [272]	Automatical Center of Lille	Vitrectomy, reconstructive Surgery	Arc RCM	7	/	/	/	/	/	Master-slave	No	Yes	No	Yes	No	Three fiber optic sensors are placed at the end of the knife to detect the object tracking surface.
Dual-arm robotic system [84,273,274,275]	Columbia Univ.	Retinal surgery	IODR + SGP	11	ϕ12 * 21	20 * 20 * 40	5	Φ84 * 150	1.022	Master-slave	No	Yes	Yes	Yes	No	Vascular cannulation and vascular stenting were accomplished in an animal mode.
MSVS [60,276]	Tokyo Univ.	Vitreoretinal Surgery	Arc RCM	6	XYZ: ±75 insert: 65	±90 * 80	>10	390 * 408 * 1058	/	Master-slave	No	Yes	No	No	No	Microvascularization experiments on pig eyeballs with successful microvascular cannulation.
SHER [6,70,277,278,279]	Johns Hopkins Univ.	Retinal surgery (ERM, RVC)	Parallel six-bar mechanism	6	±50	±30	3	/	>25	Co-manipulated	No	Yes	Yes	Yes	No	Experiments such as retinal capsulotomy and venous cannulation have been accomplished on biological models. Recently, a force sensing instrument based on FBG sensors and a position sensing instrument based on CP-SSOCT integrated at the end were developed.
Micron [52,173,182,231,280,281]	Carnegie Mellon & Johns Hopkins Univ.	Retinal surgery (ERM, RVC, laser photo-coagulation)	SGP	6	Φ 4 * 4	/	4	Φ28.5 * 126	0.07	Handheld	Yes	Yes	Yes	Yes	No	Generation 2.0 integrates electrodynamic force-sensitive microneedles, introduces a customized 2-DOF force sensor. Experiments on intravenous cannulation were performed in isolated porcine eyes.
PSS [53,55,260,261,264,265,270]	Preceyes BV	Retinal surgery (ERM, RVC)	Parallelo-gram RCM	4	insert:32	±45 * 360	<10	65 * 65 * 270	0.89	Master-slave	Yes	Yes	No	Yes	Yes	It has completed clinical trials of RVC and ERM, and has conducted numerous comparative performance trials of robotic-assisted and manual surgery by combining EYESI
IRISS [57,153,230,257,258]	UCLA	Cataract surgery, retinal surgery (RVC)	Arc RCM	7	XYZ: ±12.5 insert: 85	120	27 ± 2	/	/	Master-slave	Yes	Yes	Yes	Yes	Yes	OCT technology was integrated at IRISS for experiments in animal models of lens extraction. Recently, automated cataract extraction was evaluated in porcine eyes.
RAM!S [62,68,282,283]	TUM	Retinal surgery (RVC)	PCJM	6	±28	±58.7	1	185 * 44 * 226	0.306	Master-slave	No	Yes	Yes	Yes	Yes	Controllers with force feedback characteristics were developed to implement positional error control. Recently, subretinal depth tracking of needles using OCT was evaluated in isolated porcine eyes.
Co-manipulator [3,7,21,179,271]	Catholic University of Leuven	Retinal surgery (RVC)	Parallelo-gram RCM	4	insert:30	50 * ± 28 * 360	10	27 * 37 * 20	1.47	Master-slave	Yes	Yes	Yes	Yes	No	In 2018, four patients were treated in a clinical trial. The drug was injected into a retinal vein and successfully sustained for 10 min. Currently, efforts are underway to commercialize the developed technology.
EyeDoc [61,284,285,286,287]	Beihang Univ.	Retinal vascular bypass surgery	Parallelo-gram RCM	5	20 * 20 * 30	±45	1	/	/	Master-slave	Yes	Yes	No	No	No	Retinal photocoagulation endolaserization was accomplished on living rabbits. PPV and retinal vascular cannulation were performed on isolated porcine eyes. Recently, generation 2.0 was developed and experimented on isolated porcine eyes.
MUSA [41,51,66,255]	MICROSURE	Reconstructive surgery	Parallelo-gram with 4 bar linkages	7	Biconical ϕ60 * 45	/	70	250 * 200 * 100	8	Master-slave	Yes	Yes	No	No	No	The MSR as 1.0 generation has completed 7 anastomoses in rats. In 2020, improved to 2.0 generation MUSA and completed human clinical trials.
NeuroArm [4,152,241,243]	University of Calgary and MDA Ltd.	Neurosurgery, stereotaxy	Tandem arm	7	60 * 60 * 60	103 * 62 * 112	50	/	/	Master-slave	Yes	Yes	Yes	Yes	No	It has been used in dozens of clinical cases of microsurgery, and the next generation of the system is under development with improved structure, precision, and HMI.
REMS [30,64,244,245,247,248,250]	Galen Robotics, Inc.	Otolaryngology surgery	Parallel mechanisms	6	XYZ: 125 * 125 * 125 insert:65	Roll,Tilt,Rotation:120,60,360	Approximately 500	250 * 250 * 350	<10	Co-manipulated	No	Yes	YES	YES	No	Animal and human cadaver experiments have been conducted, and the group is currently optimizing the HMI performance of the system.

SMOS = Stereotaxical Microtelemanipulator for Ocular Surgery; IODR = Intra-Ocular Dexterity Robots; SGP = Stewart-Gough platform; MSVS = Microsurgical robotic system for vitreoretinal surgery; SHER = Steady-Hand Eye Robot; IRIS = Integrated Robotic Intraocular Snake; TUM = Technical University of Munich; PCJM = Parallel Coupled Joint Mechanism.

**Table 5 sensors-23-08503-t005:** Current challenges for developments in MSR system.

Challenges	Descriptions
Clinical acceptance	Acceptance by clinicians and patients is required, as well as by third-party payers in the health-care system, including insurance companies.
Ethical and legal concerns	The internationally recognized regulatory standards have not yet been established. And the development of surgical robots may intensify the contradiction in the allocation of medical resources.
Interdisciplinary subject	An interdisciplinary approach is required to effectively address both clinical and engineering aspects.
Human factors	Human factors considerations need to be components of the design to yield safer, more usable, and effective devices.
Human–machine interaction	Robotic surgery reduces direct surgeon–patient interaction, which places high demands on the development of effective user interfaces and intuitive feedback systems.
Visualization	Visual feedback based only on the microscope is not comprehensive, affecting the surgeon’s manipulation precision as well as the perception of the surgical environment.
Control robustness	Establishing robust control systems presents engineering challenges, including the implementation of auxiliary control functions to handle unexpected events or disturbances during surgery.
Wireless networks	The speed and stability of the wireless network directly affects the quality of the procedure. Also, the network security remains a major issue, including the protection of patient privacy.
Technical training	Well-trained physicians require extensive technical training. The development of effective training simulators is crucial to address this challenge.

## Data Availability

Not applicable.

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
