# Peer review of "Microsurgery Robots: Applications, Design, and Development"

_sensors, 2023, doi:10.3390/s23208503_

Round 1

Reviewer 1 Report

Review Microsurgery Robots: Applications, Design and Development is in subject of Sensors journal

This valuable review presented  technical challenges in microsurgery robots (MSR), and systematically summarizes the key technologies in MSR for different methods of robots control and human-machine interaction, and further to the ability in achieving a certain level of autonomy.

Paper good written based on very large number of references (298), which are detail used and described.

The structure of article is relevant to presented idea. Paper need minor review for delete some editorial mistakes. Conclusion need improve.

In review please consider my remark presented in few comments.

Comment 1

You present Table 1 (page 6/253)

Table 1. Examples of clinical specialties requiring microsurgery.

before it citation in Subsection

2.5. Summary  (page 6

Table 1 provides a comprehensive summary of the technical parameters and challenges associated with microsurgical tasks in different specialties.

I propose move Table 1 after this sentence.

Comment 2

You wrote using numbering

Overall, the challenges 252

in microsurgery can be mainly categorized into the following aspects:

1. Microsurgery involves the manipulation of micron-scale targets, including the treatment of delicate and fragile tissues (such as epiretinal membranes or brain tissues), as well as suturing or injecting small vessels, nerves, and lymphatic channels. These tasks require a high degree of precision, and inadvertent tremors as the surgeon manipulates the instruments can compromise accuracy and potentially damage the targets.

2. The surgeon's perception of the surgical environment is limited in microsurgery. The limited field of view and depth of field of the surgical microscope makes it difficult to perceive the position and depth information of small or transparent targets, and the subtle interaction forces during the surgical procedure can sometimes go unnoticed by the surgeon.

3. Microsurgical procedures require surgeons to maintain a high level of concentration, often performing prolonged surgical tasks in ergonomically unfavorable positions. This can lead to physical and mental fatigue, increasing the risk of inadvertent errors.

4. Due to the precision and complexity of microsurgical tasks, surgeons require extensive professional training before performing clinical procedures.

Because numbering was used for sections I propose use for this part bullet list

in microsurgery can be mainly categorized into the following aspects:

·         Microsurgery involves the manipulation of micron-scale targets, including the treatment of delicate and fragile tissues (such as epiretinal membranes or brain tissues), as well as suturing or injecting small vessels, nerves, and lymphatic channels. These tasks require a high degree of precision, and inadvertent tremors as the surgeon manipulates the instruments can compromise accuracy and potentially damage the targets.

·         The surgeon's perception of the surgical environment is limited in microsurgery…….

That same for

Based on the workflow of the MSR system illustrated in Figure 2, this chapter divides the key technologies of the MSR into four sections, which will be introduced in subsequent sections as follows:

1. Operation modes & mechanism designs: As the foundation of the robot system, this section discusses the structural design and control types of the MSR.

2. Sensing & perception: This is the medium through which surgeons and robots perceive the surgical environment, and this section discusses the techniques that used in MSR systems to collect environmental data.

3. Human-machine interaction (HMI): This section focuses on the interaction and collaboration between the surgeon and MSR, discussing techniques that can improve surgeon precision and comfort, as well as provide more intuitive feedback on surgical information.

4. Automation: This section discusses technologies for the robot to automatically or semi automatically perform surgical tasks, which can improve surgical efficiency and reduce the workload of the surgeon.

Please also change numbering  on bullet list for this part.

Please check whole paper and change numbering on bullet list in similar situations.

Comment 3

You wrote

There is a category according to the degree of robotic control from low to high, as Figure 3 shown.

[ Figure 3]

Figure 3. Types of surgical robots.

Shown is a category that encapsulates main types of robots with different levels of control.

 (a). Handheld robot. In the handheld robotic system, the surgical tool itself is retrofitted into a miniature robotic system called robotic tool. The surgeon manipulates it to perform the surgical procedure. The robotic tool provides tremor elimination, depth locking and other functions. "Micron" is a typical example [49].

(b). Teleoperated robot. In the teleoperated robotic system, the surgeon manipulates…….

Please consider to change this description on

There is a category according to the degree of robotic control from low to high, as Figure 3 shown a category that encapsulates main types of robots with different levels of control:

·         Handheld robot (a),

·         Teleoperated robots (b),

·         Co-manipulated robot (c),

·         Partially automated robot (d).

 [ Figure 3]

Figure 3. Types of surgical robots. (a) Handheld robot, (b)Teleoperated robots, (c) Co-manipulated robot, (d) Partially automated robot.

****** I propose to change description of Figure 3 simmlary to your description of Figure 4 ****

Handheld robot. In the handheld robotic system, the surgical tool itself is retrofitted into a miniature robotic system called robotic tool. The surgeon manipulates it to perform the surgical procedure. The robotic tool provides tremor elimination, depth locking and other functions. "Micron" is a typical example [49].

 Teleoperated robot. In the teleoperated robotic system, the surgeon manipulates master module to control slave module which replaces the surgeon's hand to manipulate the surgical tool. The system integrates the functions of motion scaling and tremor filtering through servo algorithms. In addition, it achieves three-dimensional perception with the integration of haptic feedback or depth perception algorithms at the end of the surgical tool. A typical example is "Preceyes Surgical System" [50,51].

Co-manipulated robot. In the co-manipulated robotic system, the surgeon manipulates the surgical tool simultaneously with the robot. The surgeon manually manipulates the surgical tool directly to control the motion. At the same time, the robot also holds the surgical tool, which provides assistive compensation for hand tremor and allows for prolonged immobilization of the surgical tool. "Co-manipulator" is a typical example [52].

Partially automated robot. In the partially automated robotic system, specific procedures or steps of procedures are performed automatically by the robot. The robot directly manipulates and controls the motion of the surgical tool. The processed image information is provided to the robot as feedback and guidance. Simultaneous visual information is transmitted to the surgeon, who can provide override orders to supervise the partially automated procedure at any time. "IRISS" is a typical example [53].

Comment  4

You wrote

The RCM mechanism is divided into passive joint, active control and mechanical constraint according to the implementation method.

(a). Passive-joint RCM mechanism. It is generally composed of two degrees of freedom vertically intersecting rotary joints, which achieve RCM through the active jointmovement of the robotic arm under the restriction of the incision on the patient's body surface. This design guarantees safety while reducing number of joints and mechanism size. However, it is easy to be influenced by the flexibility at the human incision and difficult to determine the accurate position of instrument insertion, which has a bad impact on the manipulation of precision. "MicroHand S" is a typical example [61].

(b). Active-control RCM mechanism. In this type of RCM mechanism, the RCM of the surgical tools around the incision is generally achieved by software control algorithm, which is usually called virtual RCM. This design is simple in structure and flexible in form. However, the precision of the movement of the surgical tools depends on the stability of the precision of the control system. The general concern is that the security of the system is guaranteed by the algorithm. "RAMS" is a typical example [66].

(c). Mechanical-constraint RCM mechanism. It generally used specific mechanical mechanism to achieve the RCM of surgical tools, ………

I propose change this list on bullet

The RCM mechanism is divided into passive joint, active control and mechanical constraint according to the implementation method.

·         Passive-joint RCM mechanism. It is generally composed of two degrees of freedom vertically intersecting rotary joints, which achieve RCM through the active jointmovement of the robotic arm under the restriction of the incision on the patient's body surface. This design guarantees safety while reducing number of joints and mechanism size. However, it is easy to be influenced by the flexibility at the human incision and difficult to determine the accurate position of instrument insertion, which has a bad impact on the manipulation of precision. "MicroHand S" is a typical example [61].

·         Active-control RCM mechanism. In this type of RCM mechanism, the RCM of the surgical tools around the incision is generally achieved by software control algorithm, which is usually called virtual RCM. This design is simple in structure and flexible in form. However, the precision of the movement of the surgical tools depends on the stability of the precision of the control system. The general concern is that the security of the system is guaranteed by the algorithm. "RAMS" is a typical example [66].

·         Mechanical-constraint RCM mechanism. It generally used specific mechanical mechanism to achieve the RCM of surgical tools, ………

Comment  5

Please move citation of Table 3 in context of article

“Table 3 briefly describes the main points of depth perception methods along with their corresponding references”  (page 15/588)

 before Table 3 (page 14.573)

Comment  6

In bullet list from (page 18/780) you use other type list

Some primary advantages of virtual fixtures include: (i) the ability to easily define and modify fixture characteristics and dynamics within the software (such as stiffness, coarse, and viscous environment), (ii) the absence of mass or mechanical constraints, and the elimination of maintenance needs, and (iii) the capability to easily develop, customize, and adapt these fixtures based on the surgical corridor for a specific patient [4,183].

I propose  use of next level graphical bullet list

Some primary advantages of virtual fixtures include:

Ø  the ability to easily define and modify fixture characteristics and dynamics within the software (such as stiffness, coarse, and viscous environment),

Ø  the absence of mass or mechanical constraints, and the elimination of maintenance needs,

Ø  the capability to easily develop, customize, and adapt these fixtures based on the surgical corridor for a specific patient [4,183].

Comment  7

Conclusions look too short for your article. I propose to describe more detail your observation based on very rich number of references presented in article.  You can also add remark about possibility of increase of MSR autonomy in future designs based on artificial intelligence.

I propose use for improved conclusions bullet list it improve readability of this part

Author Response

Comments and Suggestions for Authors

Review Microsurgery Robots: Applications, Design and Development is in subject of Sensors journal

This valuable review presented technical challenges in microsurgery robots (MSR), and systematically summarizes the key technologies in MSR for different methods of robots control and human-machine interaction, and further to the ability in achieving a certain level of autonomy.

Paper good written based on very large number of references (298), which are detail used and described.

The structure of article is relevant to presented idea. Paper need minor review for delete some editorial mistakes. Conclusion need improve.

In review please consider my remark presented in few comments.

We appreciate the reviewer’s detailed comments which have been helpful in improving our manuscript. We have addressed the concerns brought up and included the recommended revision highlighted in the revised version.

Comment 1

You present Table 1 (page 6/253)

Table 1. Examples of clinical specialties requiring microsurgery.

before it citation in Subsection

2.5. Summary  (page 6

Table 1 provides a comprehensive summary of the technical parameters and challenges associated with microsurgical tasks in different specialties.

I propose move Table 1 after this sentence.

We apologize for the oversight in the placement of Table 1 in the manuscript. Accordingly, we have made the necessary adjustment and moved Table 1 to the suggested location.

The amendments are highlighted and can be found in Section 2.5.

Comment 2

You wrote using numbering

Overall, the challenges 252

in microsurgery can be mainly categorized into the following aspects:

  1. Microsurgery involves the manipulation of micron-scale targets, including the treatment of delicate and fragile tissues (such as epiretinal membranes or brain tissues), as well as suturing or injecting small vessels, nerves, and lymphatic channels. These tasks require a high degree of precision, and inadvertent tremors as the surgeon manipulates the instruments can compromise accuracy and potentially damage the targets.
  2. The surgeon's perception of the surgical environment is limited in microsurgery. The limited field of view and depth of field of the surgical microscope makes it difficult to perceive the position and depth information of small or transparent targets, and the subtle interaction forces during the surgical procedure can sometimes go unnoticed by the surgeon.
  3. Microsurgical procedures require surgeons to maintain a high level of concentration, often performing prolonged surgical tasks in ergonomically unfavorable positions. This can lead to physical and mental fatigue, increasing the risk of inadvertent errors.
  4. Due to the precision and complexity of microsurgical tasks, surgeons require extensive professional training before performing clinical procedures.

Because numbering was used for sections I propose use for this part bullet list

in microsurgery can be mainly categorized into the following aspects:

  • Microsurgery involves the manipulation of micron-scale targets, including the treatment of delicate and fragile tissues (such as epiretinal membranes or brain tissues), as well as suturing or injecting small vessels, nerves, and lymphatic channels. These tasks require a high degree of precision, and inadvertent tremors as the surgeon manipulates the instruments can compromise accuracy and potentially damage the targets.
  • The surgeon's perception of the surgical environment is limited in microsurgery…….

That same for

Based on the workflow of the MSR system illustrated in Figure 2, this chapter divides the key technologies of the MSR into four sections, which will be introduced in subsequent sections as follows:

  1. Operation modes & mechanism designs: As the foundation of the robot system, this section discusses the structural design and control types of the MSR.
  2. Sensing & perception: This is the medium through which surgeons and robots perceive the surgical environment, and this section discusses the techniques that used in MSR systems to collect environmental data.
  3. Human-machine interaction (HMI): This section focuses on the interaction and collaboration between the surgeon and MSR, discussing techniques that can improve surgeon precision and comfort, as well as provide more intuitive feedback on surgical information.
  4. Automation: This section discusses technologies for the robot to automatically or semi automatically perform surgical tasks, which can improve surgical efficiency and reduce the workload of the surgeon.

Please also change numbering on bullet list for this part.

Please check whole paper and change numbering on bullet list in similar situations.

We agree that the numbering of the above cases needs to be adjusted to avoid confusion. We have revised numbering on bullet list in Section 2.5, and changed the numbering in section 3.1 to "â‘  â‘¡ â‘¢...... " format, as this symbol corresponds to the annotation in the Figure 2.

The amendments can be found in Section 2.5 & 3.1.

Comment 3

You wrote

There is a category according to the degree of robotic control from low to high, as Figure 3 shown.

[ Figure 3]

Figure 3. Types of surgical robots.

Shown is a category that encapsulates main types of robots with different levels of control.

 (a). Handheld robot. In the handheld robotic system, the surgical tool itself is retrofitted into a miniature robotic system called robotic tool. The surgeon manipulates it to perform the surgical procedure. The robotic tool provides tremor elimination, depth locking and other functions. "Micron" is a typical example [49].

(b). Teleoperated robot. In the teleoperated robotic system, the surgeon manipulates…….

Please consider to change this description on

There is a category according to the degree of robotic control from low to high, as Figure 3 shown a category that encapsulates main types of robots with different levels of control:

  • Handheld robot (a),
  • Teleoperated robots (b),
  • Co-manipulated robot (c),
  • Partially automated robot (d).

 [ Figure 3]

Figure 3. Types of surgical robots. (a) Handheld robot, (b)Teleoperated robots, (c) Co-manipulated robot, (d) Partially automated robot.

****** I propose to change description of Figure 3 simmlary to your description of Figure 4 ****

Handheld robot. In the handheld robotic system, the surgical tool itself is retrofitted into a miniature robotic system called robotic tool. The surgeon manipulates it to perform the surgical procedure. The robotic tool provides tremor elimination, depth locking and other functions. "Micron" is a typical example [49].

 Teleoperated robot. In the teleoperated robotic system, the surgeon manipulates master module to control slave module which replaces the surgeon's hand to manipulate the surgical tool. The system integrates the functions of motion scaling and tremor filtering through servo algorithms. In addition, it achieves three-dimensional perception with the integration of haptic feedback or depth perception algorithms at the end of the surgical tool. A typical example is "Preceyes Surgical System" [50,51].

Co-manipulated robot. In the co-manipulated robotic system, the surgeon manipulates the surgical tool simultaneously with the robot. The surgeon manually manipulates the surgical tool directly to control the motion. At the same time, the robot also holds the surgical tool, which provides assistive compensation for hand tremor and allows for prolonged immobilization of the surgical tool. "Co-manipulator" is a typical example [52].

Partially automated robot. In the partially automated robotic system, specific procedures or steps of procedures are performed automatically by the robot. The robot directly manipulates and controls the motion of the surgical tool. The processed image information is provided to the robot as feedback and guidance. Simultaneous visual information is transmitted to the surgeon, who can provide override orders to supervise the partially automated procedure at any time. "IRISS" is a typical example [53].

We appreciate the reviewer’s suggestions. We have structured the caption of Figure 3 with descriptive categories like that of Figure 4 as advised, and the associated description has also been modified.

The amendments can be found in Section 3.2.

Comment  4

You wrote

The RCM mechanism is divided into passive joint, active control and mechanical constraint according to the implementation method.

(a). Passive-joint RCM mechanism. It is generally composed of two degrees of freedom vertically intersecting rotary joints, which achieve RCM through the active jointmovement of the robotic arm under the restriction of the incision on the patient's body surface. This design guarantees safety while reducing number of joints and mechanism size. However, it is easy to be influenced by the flexibility at the human incision and difficult to determine the accurate position of instrument insertion, which has a bad impact on the manipulation of precision. "MicroHand S" is a typical example [61].

(b). Active-control RCM mechanism. In this type of RCM mechanism, the RCM of the surgical tools around the incision is generally achieved by software control algorithm, which is usually called virtual RCM. This design is simple in structure and flexible in form. However, the precision of the movement of the surgical tools depends on the stability of the precision of the control system. The general concern is that the security of the system is guaranteed by the algorithm. "RAMS" is a typical example [66].

(c). Mechanical-constraint RCM mechanism. It generally used specific mechanical mechanism to achieve the RCM of surgical tools, ………

I propose change this list on bullet

The RCM mechanism is divided into passive joint, active control and mechanical constraint according to the implementation method.

  • Passive-joint RCM mechanism. It is generally composed of two degrees of freedom vertically intersecting rotary joints, which achieve RCM through the active jointmovement of the robotic arm under the restriction of the incision on the patient's body surface. This design guarantees safety while reducing number of joints and mechanism size. However, it is easy to be influenced by the flexibility at the human incision and difficult to determine the accurate position of instrument insertion, which has a bad impact on the manipulation of precision. "MicroHand S" is a typical example [61].
  • Active-control RCM mechanism. In this type of RCM mechanism, the RCM of the surgical tools around the incision is generally achieved by software control algorithm, which is usually called virtual RCM. This design is simple in structure and flexible in form. However, the precision of the movement of the surgical tools depends on the stability of the precision of the control system. The general concern is that the security of the system is guaranteed by the algorithm. "RAMS" is a typical example [66].
  • Mechanical-constraint RCM mechanism. It generally used specific mechanical mechanism to achieve the RCM of surgical tools, ………

Thank you for the valuable suggestions, we have changed the above list in Section 3.2 to bullet format.

The amendments can be found in Section 3.2.

Comment  5

Please move citation of Table 3 in context of article

“Table 3 briefly describes the main points of depth perception methods along with their corresponding references”  (page 15/588)

 before Table 3 (page 14.573)

We apologize for the misplacement of Table 3 in the manuscript. We have adjusted the description related to Table 3 to make its placement reasonable.

The amendments can be found in Section 3.3.2.

Comment  6

In bullet list from (page 18/780) you use other type list

Some primary advantages of virtual fixtures include: (i) the ability to easily define and modify fixture characteristics and dynamics within the software (such as stiffness, coarse, and viscous environment), (ii) the absence of mass or mechanical constraints, and the elimination of maintenance needs, and (iii) the capability to easily develop, customize, and adapt these fixtures based on the surgical corridor for a specific patient [4,183].

I propose  use of next level graphical bullet list

Some primary advantages of virtual fixtures include:

Ø  the ability to easily define and modify fixture characteristics and dynamics within the software (such as stiffness, coarse, and viscous environment),

Ø  the absence of mass or mechanical constraints, and the elimination of maintenance needs,

Ø  the capability to easily develop, customize, and adapt these fixtures based on the surgical corridor for a specific patient [4,183].

After careful consideration of your constructive suggestions, we have decided to further improve the readability by expressing in a concise sentence without the use of any list type.   Since the list of advantages is not meant to be exhaustive, we hope to focus on substantiating the motivation for virtual fixtures with a few representative examples. 

The amendments can be found in Section 3.4.2.

Comment  7

Conclusions look too short for your article. I propose to describe more detail your observation based on very rich number of references presented in article.  You can also add remark about possibility of increase of MSR autonomy in future designs based on artificial intelligence.

I propose use for improved conclusions bullet list it improve readability of this part

We thank the reviewer for bringing up an interesting subject relevant to the future of MSR development. We have since included some insights on the potential role of artificial intelligence in advancing the level of autonomy in MSR through further reinforcing the discussion in a dedicated section on Higher Level of Autonomy under Future directions. This content inclusion is mentioned in the conclusion section. Through careful consideration of the reviewer’s suggestion to use bullet list for readability, we organize several discussions, which the conclusion is eventually based on, in this recommended way. However, for readability of the conclusion, we hope to maintain a concise presentation of the conclusive remarks in proper sentences within paragraphs.

The amendments can be found in Section 6.

Reviewer 2 Report

Greetings to the authors, paper with an elaborate review of microsurgery robots. This is a nice presentation regarding applications versus design and development.

Appreciation

1. This paper presents the MSR for ophthalmology, neurosurgery, reconstructive surgery, urology, and otolaryngology, along with Tables 1 and 2.5 summary.

2. Interesting and technology are presented in Section 3, along with figure 2 and 3. It deals with varying sensing and perception in MSR to achieve localization, along with target detection, depth perception, and haptic sensing.

3. Human-machine interaction, feedback mechanisms, and automation are key players in MSR.

4. Classical MRS is also well-classified with respect to its kinematics.

5. Rightly mentioned the current challenges in MRS, which provide future research contributions.

Queries

The following points were not addressed in this review.

1. Computation methods or devices that are part of automation and HMI are not mentioned. provides some details.

2. The other challenge is MSR working with human life, and ethics and legal-related concerns are not mentioned.

3. The learning curve of the surgeons versus autonomous robot-related discussions was mentioned very less.

4. Kindly incorporate the MSR trade-offs in section 5.

Author Response

Reviewer: 2

Comments and Suggestions for Authors

Greetings to the authors, paper with an elaborate review of microsurgery robots. This is a nice presentation regarding applications versus design and development.

Appreciation

  1. This paper presents the MSR for ophthalmology, neurosurgery, reconstructive surgery, urology, and otolaryngology, along with Tables 1 and 2.5 summary.
  2. Interesting and technology are presented in Section 3, along with figure 2 and 3. It deals with varying sensing and perception in MSR to achieve localization, along with target detection, depth perception, and haptic sensing.
  3. Human-machine interaction, feedback mechanisms, and automation are key players in MSR.
  4. Classical MRS is also well-classified with respect to its kinematics.
  5. Rightly mentioned the current challenges in MRS, which provide future research contributions.

We thank Reviewer 2 for the positive comments and constructive suggestions.

The reviewer’s understanding of the contribution of our work is accurate. The specific lists of points to be addressed are also timely pointers in revising our manuscript. All points have been addressed in the response and the corresponding revisions are highlighted in the resubmitted manuscript.

We are also grateful for the reviewer’s encouraging remarks on the organization and quality of the work.

Queries

The following points were not addressed in this review.

  1. Computation methods or devices that are part of automation and HMI are not mentioned. provides some details.

In the revised manuscript, we have further expounded on the computation methods for HMI and automation such as variable admittance control and reinforcement learning, respectively. This addition will enhance the clarity and completeness of our work. Section 4 covers the aspects related to automation and HMI based on various classes of existing MSR systems such as the automation features in IRISS and HMI features in neuroArm System, which we hope to address the reviewer’s enquiry about the device aspect.

The amendments can be found in Section 3.4 & 3.5.

  1. The other challenge is MSR working with human life, and ethics and legal-related concerns are not mentioned.

The reviewer has pointed out an interesting discussion on the non-technical aspects of challenges in MSR associated with ethics and legal-related issues. These are important considerations in the process of advancing MSR technology. We have included a discussion on these aspects as advised.

The amendments can be found in Section 5.1.

  1. The learning curve of the surgeons versus autonomous robot-related discussions was mentioned very less.

Thank you for providing valuable suggestions. The learning curve of surgeons versus autonomous robots is indeed an important aspect to consider. We have conducted further research and included a discussion about this topic in the Section 2.5 of the article.

The amendments can be found in Section 2.5.

  1. Kindly incorporate the MSR trade-offs in section 5.

Thanks for the suggestions. We have expanded the discourse on trade-offs of the technology in our revised manuscript to provide readers with a more balanced understanding of the benefits and drawbacks associated with MSR systems.

The amendments can be found in Section 5.1.
